# How antisense transcripts can evolve to encode novel proteins

Bharat Ravi Iyengar [1] ✉, Anna Grandchamp[1,3] & Erich Bornberg-Bauer [1,2]

Protein coding features can emerge de novo in non coding transcripts, resulting in emergence of new protein coding genes. Studies across many species show that a large fraction of evolutionarily novel non-coding RNAs have an antisense overlap with protein coding genes. The open reading frames (ORFs) in these antisense RNAs could also overlap with existing ORFs. In this study, we investigate how the evolution an ORF could be constrained by its overlap with an existing ORF in three different reading frames. Using a combination of mathematical modeling and genome/transcriptome data analysis in two different model organisms, we show that antisense overlap can increase the likelihood of ORF emergence and reduce the likelihood of ORF loss, especially in one of the three reading frames. In addition to rationalising the repeatedly reported prevalence of de novo emerged genes in antisense transcripts, our work also provides a generic modeling and an analytical framework that can be used to understand evolution of antisense genes.

New protein coding genes often arise from existing protein coding genes. This process frequently involves duplication of an existing gene, and a subsequent divergence of one of the duplicated copies from the ancestral sequence[1–3]. Several studies have shown that protein coding genes can also emerge de novo, in DNA sequences that did not previously encode a protein (de novo gene emergence)[4–9]. A protein coding gene thus emerged does not inherit the DNA sequence features necessary for gene expression (transcription and translation), from an ancestral protein coding gene. It must therefore acquire them through random mutations.

The most basic requirement for translation is an open reading frame (ORF), which is the region of an RNA that is translated into a protein sequence. Efficient translation often requires additional features such as Kozak consensus sequences[10–12], an optimal codon usage[13], and other context dependent regulatory features present in the 5′ and 3′ untranslated regions of the RNA[14,15].

Because heritable (germline) mutations are rare in most organisms (less than 1 mutation in 100 million base pairs of DNA per generation)[16–18], it is unlikely for many features to emerge simultaneously. That is, features must evolve sequentially. This in turn means that emergence of a phenotype, such as gene expression, is more likely when some required features already exist, and the missing features emerge via mutations. For example, de novo emergence is more likely when an ORF is already present and transcriptional features emerge subsequently, or vice versa. In our recent work, we also show that de novo emergence is more likely via the trajectory where transcription emerges before the emergence of an ORF[19]. Thus stably synthesized RNAs that are not actively and specifically involved in protein synthesis (such as long non-coding RNAs or lncRNAs) can be good sources of new proteins.

Experimental analyses of the ribosome's footprint on RNAs (ribosome profiling) suggest that some ORFs present in lncRNAs are actively translated[20–24]. Proteins synthesized from the translation of such ORFs can also be beneficial to the host organism[22,24]. Many lncRNA genes share their genomic location with other genes, but are transcribed in the opposite direction (antisense overlap)[25–29]. A recent study has characterized previously unknown RNAs in different species of yeasts, and has shown that a large proportion of these RNA genes have an antisense overlap with existing genes[23]. This study also shows that ORFs contained in these RNAs show signatures of translation. These translated ORFs also include those that have recently emerged in one specific species of yeast. However, these species-specific ORFs

[1]Institute for Evolution and Biodiversity, University of Münster, Hüfferstrasse 1, Münster, Germany. [2]Department of Protein Evolution, Max Planck Institute for Biology Tübingen, Max-Planck-Ring 5, Tübingen, Germany. [3]Present address: Aix-Marseille Université, INSERM, TAGC, Marseille, France. ✉e-mail: b.ravi@uni-muenster.de

**Table 1 | Mutation bias probabilities for different nucleotide mutations in *Saccharomyces cerevisiae*[17]**

| Substitution | Probability ($\mu$) |
|---|---|
| A:T → T:A | 0.063 |
| A:T → G:C | 0.144 |
| A:T → C:G | 0.110 |
| G:C → A:T | 0.349 |
| G:C → T:A | 0.182 |
| G:C → C:G | 0.152 |

A:T denotes an A-T base pair in a double-stranded DNA. Thus A → G mutation on one DNA strand would cause a T → C mutation on the complementary strand. We describe the other mutations in the same way. For our model, we used the reported mutation rate of $1.7 \times 10^{-10}$ mutations per nucleotide position per generation, in diploid *Saccharomyces cerevisiae* cells[17]. For mutation bias probabilities in *D. melanogaster*, see Table S1.

are less efficiently translated than the ORFs that are conserved between different species. Overall, this study lends support to a hypothesis that many new proteins arise from antisense RNAs. It is likely that the ORFs encoding such proteins are also antisense to existing genes.

In this study, we analyse the emergence of ORFs in antisense RNAs. We specifically focus on ORFs that have an antisense overlap with the coding region (canonical ORF) of an existing protein coding gene. We refer to these ORFs as antisense ORFs (asORFs). Evolution of asORFs is also interesting because it is constrained by the evolutionary selection pressure on the overlapping protein coding genes[30,31]. A pair of mutually antisense ORFs can overlap with each other in three different reading frames. That is, the codon positions in the two ORFs can either perfectly overlap or be offset by one or two nucleotides. The constraints on the co-evolution of the two ORFs would be different in the different reading frames[31]. Our study aims to explore the constraints that affect the evolution of asORFs. To this end, we employ a mathematical model to calculate the probabilities of asORF emergence and loss, in each of the three reading frames. Using the model, we predict that one of the reading frames has a higher propensity to harbour ORFs. We also predict that the likelihood of ORF emergence in this reading frame is higher, and that of ORF loss is lower, than in the other two reading frames. We support our model's predictions with genome analysis of two different organisms—*Saccharomyces cerevisiae* and *Drosophila melanogaster*. We also find that emergence of asORFs in reading frame 1 can be more likely than emergence of non-antisense (intergenic) ORFs.

## Results

We developed a mathematical model to estimate the probabilities of ORF emergence and loss, in DNA regions antisense to existing protein coding ORFs. This model is defined by two kinds of probability. The first is the probability of finding a certain kind of DNA sequence, for example an ORF. This stationary probability depends on the nucleotide composition of the DNA region that can be roughly approximated by GC content or by the frequencies of short DNA sequences (oligomers). The second kind of probability describes the mutational change of a sequence to a different kind of sequence. For example, gain or loss of an ORF. This transition probability depends on the mutation rate and mutation bias, in addition to nucleotide composition. We estimate these parameters primarily from the data on the yeast, *Saccharomyces cerevisiae* (Table 1)[17]. Our choice is motivated by the fact that the budding yeast is a convenient model organism for laboratory experimental studies that can be used to validate several of our theoretical predictions. We also performed analogous analyses using data obtained from *Drosophila melanogaster* (Table S1)[16].

We estimated the stationary and transition probabilities of antisense ORFs (asORFs, Equations (1)–(3)) using the existing (sense) ORF as a reference. asORFs can overlap with the sense ORFs in three

different reading frames (henceforth referred to as just "frames"). In frame 0, the codons in the asORF exactly overlap the codons in the sense ORF. In frames 1 and 2, the codons in the asORF are shifted towards the 5′ end of the sense ORF by one and two nucleotide positions, respectively. Thus in frames 1 and 2, the sequence of an antisense codon is determined by two partially overlapping sense codons (dicodons, Fig. 1A). Due to this sequence overlap, the evolution of asORFs would be constrained by the evolutionary selection pressures on the sense ORF. Furthermore, these constraints would be different for asORFs located in the three different frames. We analysed the evolution of asORFs when the sense ORF is under three different levels of purifying selection, defined in our study as follows. The first level describes an absence of purifying selection, where any kind of mutation except a nonsense mutation (gain of stop codon) in the sense ORF is tolerated. The second level describes a weak purifying selection that allows synonymous mutations, as well as mutations where an amino acid is substituted by a chemically similar amino acid (for example, aspartic acid to glutamic acid; see 'Methods'). Finally, the third level describes a strong purifying selection, where only synonymous mutations are tolerated in the sense ORF.

### Antisense ORFs are generally more likely to exist in frame 1

For any stretch of DNA to be an ORF, its sequence should contain $3n$ nucleotides ($n \geq 3$), with a start codon that marks its beginning, and exactly one stop codon that marks its end. The absence of any stop codon within the DNA sequence is the most important factor in determining the existence of an ORF. That is because the likelihood of a premature stop codon increases exponentially with the ORF's length, whereas the likelihoods of a start codon and a terminal stop codon are independent of the ORF's length (Equations (1)–(3)).

Based on these considerations, we determined the probability of finding an asORF of a given length. To this end, we first calculated the probability of finding a stop codon in the three antisense frames (antisense stop codons), given the condition that no (sense) stop codon exists within the overlapping sense ORF. An antisense stop codon can exist in frame 0 wherever the three reverse complementary codons (CTA, TTA, TCA) exist in the sense ORF. Because these three codons are allowed in the sense ORF, the overlap does not affect the antisense stop codon's probability in frame 0. An antisense stop codon in frames 1 or 2, overlaps with a dicodon in the sense ORF (Fig. 1A). While three positions in the dicodon are determined by the antisense stop codon, the other three positions can contain any of the four nucleotides. Therefore, there are $3 \times 4^3 = 192$ possible dicodons that overlap with the three antisense stop codons. However, this set of overlapping 192 dicodons is not identical for antisense stop codons in frames 1 and 2. Specifically, 64 out of 198 dicodons that overlap an antisense stop codon in frame 1, contain a stop codon and cannot exist in the sense ORF by definition. Therefore, the number of possible dicodons that overlap an antisense stop codon in frame 1 reduces to 128. The probability of finding an antisense stop codon in frame 1, is equivalent to the probability of finding the 128 allowed dicodons. In contrast, antisense stop codons in frame 2 can overlap with all the possible 192 dicodons, and their probabilities are thus unaffected by the overlap (see Supplementary Section 2). In other words, the probability of an antisense stop codon in frame 2 is only determined by its three nucleotide positions as in case of frame 0. Codon and dicodon probabilities depend on the nucleotide composition, which can be approximated by the GC content of the locus[19]. We calculated the probability of a start codon without considering the effect of antisense overlap because this effect would be small in magnitude. Using the start and stop codon probabilities, we estimated the probability of finding an asORF of different lengths in each of the three frames. We did so for four different values of GC content (30, 40, 50 and 60%). The probabilities of asORFs in frames 0 and 2 are identical for all lengths and GC content because the probability of antisense stop codon in

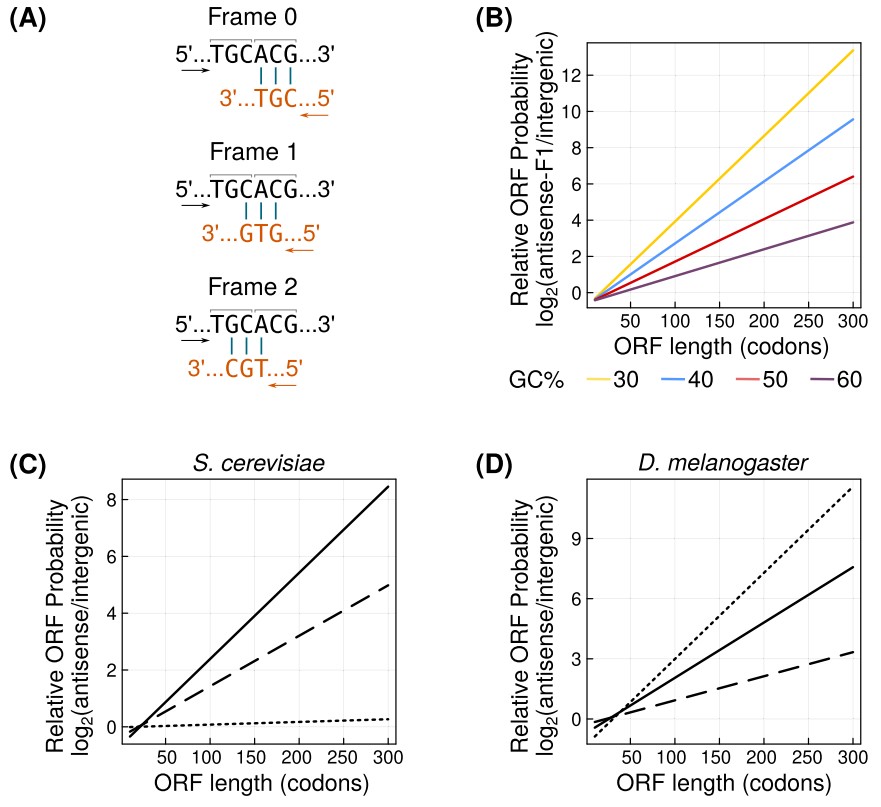

**Fig. 1 | asORFs are more likely to exist than igORFs of identical lengths and composition. A** A hypothetical antisense codon (bottom sequence, orange) can overlap with sense ORF (top sequence, black) in three different frames. Arrows indicate the direction of translation and vertical bars indicate base complementarity. Adjacent codons in the sense ORF are demarcated with horizontal square brackets. **B** The probability of asORFs in frame 1 relative to that of igORFs (log$_2$ ratio, vertical axis), for different values of GC content of the ORFs (line colours yellow = 30%, blue = 40%, red = 50%, purple = 60%). We do not show asORFs in both these frames is unaffected by the overlap. This in turn, means that asORFs in these frames are equally probable as intergenic ORFs (igORFs) with identical length and GC content. This is not the case for frame 1, where we found that asORFs are more likely to be found than in the other two frames and intergenic regions (Fig. 1B). The only exceptions are ORFs shorter than 17, 21, 27 and 39 codons present in a DNA region with a GC content of 30%, 40%, 50% and 60%, respectively. Even for these exceptional cases, the probability of an asORF in frame 1 is no less than 74% of the corresponding ORF probabilities in the other frames. More generally, the overall probability of finding an ORF of any length between 10 to 300 codons and any GC content between 30% to 60%, is higher in frame 1 than in the other two frames. We expect that igORFs can indeed be more numerous than asORFs if intergenic regions are long. Our results merely suggest that given that length and GC content are identical, the probability of an ORF increases when it has an antisense overlap with an existing ORF in frame 1.

We also calculated the probability of asORFs using actual codon and dicodon frequencies in annotated yeast ORFs. Likewise, we calculated the probability of igORFs using the frequencies of DNA trimers in yeast intergenic genome. With this analysis, we found that asORFs longer than 17, 21, and 19 codons, in frames 0, 1 and 2, respectively, are more likely to exist than igORFs of the same lengths (Fig. 1C).

The probability of finding an ORF does not depend on mutation rate bias. Therefore, ORF probability calculations using GC content (Fig. 1B) is organism-independent. However, when we computed the ORF probabilities using the frequencies of codons, dicodons and

frames 0 and 2 because their probabilities are identical to that of igORFs. The probability of asORFs relative to that of igORFs (log$_2$ ratio, vertical axis), calculated using frequencies of short DNA sequences from **C** the yeast genome, and **D** the fruitfly genome. Frames 0, 1 and 2 are denoted by dotted, solid and dashed lines, respectively. Horizontal axes in panels (**B**)–(**D**) show the length of the ORFs. We only show asORFs that overlap completely with the sense ORF. Source data are provided as a Source data file.

intergenic trimers from *D. melanogaster*, we found that frame 0 was most likely to harbour long asORFs (>38 codons; Fig. 1D). This difference between the predicted ORF probabilities of two organisms results because of differences in codon usage between the two organisms. Specifically, the codons that overlap stop codons (TTA, CTA, TCA) in antisense frame 0 encode serine and leucine. Both these amino acids are encoded by six codons each, and have similar frequencies in the coding regions of both the organisms. However, the usage of the codons—TTA, CTA, TCA, to encode the corresponding amino acids is relatively higher in *S. cerevisiae* than in *D. melanogaster* (Supplementary Section 3; Figure S1). Our GC content-based analysis (Fig. 1B) shows that probability of asORFs in frames 0 and 2 should be identical if they have the same GC content. Although mathematically valid, this is unlikely to be the case in real genomes where the nucleotide distribution cannot be approximated as a uniform distribution based on an average GC content (Fig. 1C, D).

**Antisense ORFs are frequently located in frame 1**

Our mathematical model predicts that frame 1 is more likely to harbour asORFs than the other two frames. To verify this prediction, we analysed the genome of the budding yeast, *S. cerevisiae*. We specifically chose this yeast as a model because most of its genes lack introns. This in turn allows us to investigate asORFs whose overlap with the sense ORFs is not interrupted by intronic sequences. Our choice of yeast as a model was further motivated by the availability of data on novel antisense RNAs identified in a recently published study[23]. This study

**Table 2 | Expected (using model) and observed (using *getorf*[32]) numbers of antisense and intergenic ORFs**

|  | Antisense Frame 0 | Antisense Frame 1 | Antisense Frame 2 | Intergenic |
|---|---|---|---|---|
| Total loci | 7,985,381 | 7,985,381 | 7,985,381 | 798,843,580 |
| Expected number | 592 (612) | 657 (690) | 632 (612) | 49,786 (49,646) |
| Observed number | 447 | 646 | 548 | 40,647 |
| Observed number + subORFs | 494 | 903 | 623 | 48,598 |
| Expected frequency | $7.4 \times 10^{-5}$ ($7.7 \times 10^{-5}$) | $8.2 \times 10^{-5}$ ($8.6 \times 10^{-5}$) | $7.9 \times 10^{-5}$ ($7.7 \times 10^{-5}$) | $6.2 \times 10^{-5}$ ($6.2 \times 10^{-5}$) |
| Observed frequency (+ subORFs) | $6.2 \times 10^{-5}$ | $1.1 \times 10^{-4}$ | $7.8 \times 10^{-5}$ | $6.1 \times 10^{-5}$ |

Expected numbers and frequencies of ORFs within parentheses were estimated using GC content of each locus, whereas those outside the parentheses were estimated using DNA oligomer frequencies. For both expected and observed number of asORFs, we only consider ORFs that overlap completely with a sense ORF. Here "sub-ORFs" refers to smaller ORFs (≥30 nt) that exist within an ORF such both ORFs share the same stop codon.

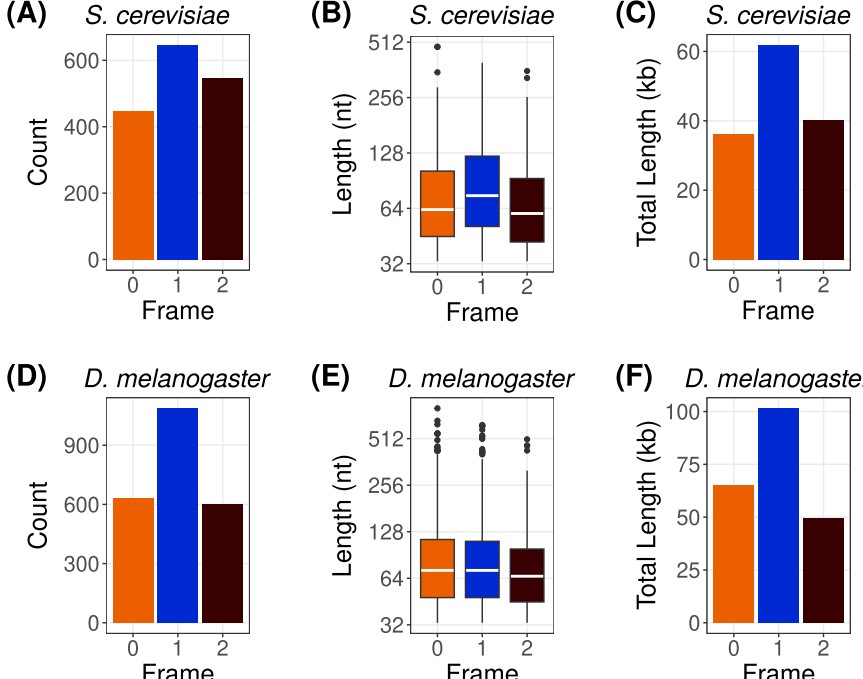

**Fig. 2 | asORFs preferentially exist in frame 1 than in the other two frames.** Total number of asORFs (vertical axis) in **A** *S. cerevisiae* and **B** *D. melanogaster*. Length distribution (vertical axes) of **B** *S.cerevisiae* (sample sizes−frame 0: 494, frame 1: 646, and frame 2: 623) and **E** *D. melanogaster* asORFs (sample sizes−frame 0: 447, frame 1: 1087, and frame 2: 548) denoted by boxplots where the boxes extend from the first to the third quartile and the whiskers have a length equal to 1.5 × the interquartile range. In each boxplot, we indicate the median length using a white horizontal bar. Cumulative length of all (**C**) *S. cerevisiae* and (**F**) *D. melanogaster* asORFs (vertical axis). In all the panels, the horizontal axes denote the three different antisense frames. We only show asORFs that overlap 100% with the sense ORF. In panels **D**–**F**, we show the unique orthologous ORFs representing orthogroups constructed from the asORFs of the seven different *D. melanogaster* lines. We show the analogous plots for the specific lines in Figure S2. Source data are provided as a Source data file.

further showed that new protein coding genes can emerge de novo from these antisense RNAs. We identified all asORFs located in the novel RNAs reported in this study, and calculated the frame in which they overlap with the annotated (sense) ORFs. We also included seven annotated yeast antisense RNAs for the identification of asORFs. Next, we calculated the number of asORFs in each of the three frames, that are at least 30 nt long and are wholly contained within the boundaries of a sense ORF. We found that asORFs in frame 1 were significantly more numerous than those in the other two frames (one-tailed Fisher exact test, FDR corrected $P < 4 \times 10^{-5}$). Specifically, ~39% of all asORFs were located in frame 1, while ~33% and ~28% asORFs were located in frames 2 and 0, respectively (Table 2, Fig. 2). We also calculated the number of ORFs that have at least 50% of their sequences overlapping in antisense with a sense ORF. This relaxation of overlap percentage did not remarkably increase the number of identified asORFs. To understand if the observed number and proportion of asORFs are in agreement with the model, we calculated the expected number of

asORFs in each frame (Equation (6)). Specifically, we estimated the total number of expected ORFs that are at least 30 nt long and are located in genomic region where antisense RNAs overlap with a known ORF. We found that the actual asORFs in the yeast genome were 1.6–24% fewer than expected (Table 2). The ORF identification tool we used (*getorf*)[32], reports the longest ORF. However, alternate start codons can exist within the ORF sequence wherever a methionine is encoded. Our model does not reject short ORFs (sub-ORFs) within a longer ORFs. When we included the sub-ORFs (≥30 nt), the observed asORFs in frame 1 were significantly more numerous than expected (one-tailed Fisher exact test, $P = 5.2 \times 10^{-8}$ with locus-specific GC content, and $P = 2.5 \times 10^{-10}$ with average oligomer frequencies; Table 2). In contrast, observed asORFs in frame 0 were significantly fewer than expected (one-tailed Fisher exact test, $P < 1.7 \times 10^{-3}$). If the observed of ORFs are significantly fewer than expected then negative selection could be an explanation. We note that our calculation of expected number of asORFs (Equation (6)) assumes that existence of ORFs in the

three different frames is independent of each other. However, presence of an ORF in any one frame can reduce the probability of ORFs in overlapping alternate frames.

The probability of finding an ORF can not only determine the expected number of ORFs, but also the length of the ORFs. Therefore, we next asked if asORFs in frame 1 are generally longer than those in the other two frames. We found that asORFs in frame 1 (median length 75 nt) were significantly longer than asORFs in frame 0 and frame 2 (median length 63 nt and 60 nt, respectively; one-tailed Mann–Whitney U test, FDR adjusted $P < 10^{-4}$; Fig. 2B). Furthermore, the cumulative length of all the asORFs in frame 1 (62 kb) was higher than that of the ORFs in frames 0 and 2 (36 kb and 40 kb, respectively; Fig. 2C).

Next, we analysed if the observed frequency of igORFs is different from that of asORFs. To this end, we calculated the observed number of igORFs including the sub-ORFs, in *S. cerevisiae* genome, using a procedure identical to that we used for identifying asORFs. We then compared the frequencies of igORFs (observed ORFs relative to total loci, Table 2) with that of each type of asORFs, and found that the frequencies of all the three types of asORFs were higher than that of igORFs (one-tailed Fisher exact test, $P < 10^{-8}$). We note again that this result does not indicate that igORFs are less likely to occur than asORFs, as we show that they are indeed more numerous than asORFs (Table 2).

We also performed a similar analysis of *D. melanogaster* genome. Specifically, we used genome and transcriptome data from inbred lines obtained from seven geographically distinct *D. melanogaster* populations[33]. We used these datasets because they contain several novel RNAs that are not annotated in the reference genome. We found that among the three antisense frames, frame 1 harboured the most number of asORFs (Fig. 2D, Figure S2). The cumulative length of all the asORFs in the frame 1 was also higher than those in the other two frames (Fig. 2E, Figure S2). This was true for all the seven lines, and also for the set of unique orthologous sequences between all the lines (orthogroups). However, asORFs in frame 1 were not generally longer than those in the other two frames (Fig. 2D, Figure S2). Specifically, the median length of asORFs in frame 0 was the highest in all populations but this difference was not statistically significant in all populations (one-tailed Mann–Whitney U test, 95% confidence interval). A possible reason for the larger median length of asORF in frame 0 could be the codon usage bias in *D. melanogaster* protein coding genes (Supplementary Section 3). We also analysed if igORFs have a higher frequency than asORFs in *D. melanogaster*. We restricted this analysis to asORFs that completely overlap with a coding exon, and themselves do not have introns. That is because different exons can antisense overlap in different frames, and one cannot attribute a specific frame to an asORF. Given these restrictions, we found that asORFs were significantly less frequent than igORFs. We speculate that this difference from *S. cerevisiae* could exist because of at least two reasons. First, in *D. melanogaster*, asORFs are ~1900× less numerous than igORFs, whereas in *S. cerevisiae*, asORFs are only 24× less numerous than igORFs. Thus, the asORFs may suffer from small sample bias. Second, our requirement of complete exon overlap causes most asORFs to be short (<24 codons), such that their probability is smaller than that of similar sized igORFs (Fig. 1D).

Our analyses of both the organisms show that asORFs are more numerous in frame 1 than expected. This is especially remarkable in *D. melanogaster* where asORFs are not even predicted to be the most abundant in frame 1 (Table S2). A possible reason for this observation could be that the composition of overlapping regions may be different from that of the known ORFs in general. To find out if this is the case, we calculated the GC content of the *D. melanogaster* protein coding exons that overlap with an antisense RNA, and compared it with the GC content of all exons. We performed this analysis for every *D. melanogaster* line. We found that the exons with overlap had a significantly

lower GC content (median ~ 0.41) relative to all exons (median ~ 0.45, one-tailed Mann–Whitney U test, FDR adjusted $P < 10^{-16}$). We found similar results with *S. cerevisiae* where the GC content of protein coding regions that overlap with an antisense RNA have a lower GC content (median 0.36) than all the protein coding regions in total (median 0.39, Mann–Whitney U test, $P < 10^{-16}$). This could at least partially explain the high frequency of asORFs in frame 1 (Fig. 1B) as their probabilities increase with decreasing GC content.

ORFs that are more likely to exist are also more likely to evolve additional protein coding features. To test if this is the case, we compared the translational efficiency of *S. cerevisiae* asORFs in different frames using ribosome profiling data[24]. We did not find any significant correlation between frame and translational efficiency of asORFs (Supplementary Section 5). However, igORFs in *S. cerevisiae* had significantly higher translational efficiency than asORFs. One possible reason is that the far more numerous igORFs can have a higher total rate of evolutionary adaptation than asORFs. We did not find any significant difference between the predicted translational efficiency (Kozak consensus sequence strength) for the different asORFs, and igORFs of *D. melanogaster*.

Overall, our genome data analyses from both organisms show frame 1 is more likely to harbour asORFs, than the other two frames.

## Antisense overlap can facilitate ORF emergence and reduce ORF loss

We next analysed how likely it is for asORFs to emerge, when they are not already present. To this end, we calculated gain probability of asORFs in each of the three frames, and under three different intensities of purifying selection. We also calculated the probability of ORF gain in the intergenic regions. We found that asORFs are less likely to emerge in frames 0 and 2 than ORFs in intergenic regions, for all ORF lengths and GC content. In contrast, long asORF in frame 1 are more likely to emerge than identically sized igORFs (Fig. 3A).

Increasing the intensity of purifying selection reduces the emergence likelihood of asORFs in all the three frames. However, long asORFs in frame 1 are still more likely to emerge than identically sized igORFs, even under strong purifying selection. Specifically, the minimum ORF length at which asORFs in frame 1 are more likely to emerge than igORFs increases with GC content and the intensity of selection. For example, in the absence of purifying selection, and at a GC content of 40%, this length is 26 codons. At the same intensity of selection, this length is 46 codons when the GC content is 60%. Under strong purifying selection and a GC content of 60%, only the asORFs longer than 108 codons are more likely to emerge than identically sized igORFs (Fig. 3A). Our analogous analysis with mutation bias parameters estimated from *D. melanogaster* produced similar results (Figure S4A).

Our analysis of ORF gain probabilities using the frequencies of DNA oligomers (codons, dicodons and intergenic trimers) also shows that asORFs are very likely to emerge in frame 1 (Fig. 3B). ORFs longer than 29, 59 and 68 codons are more likely to emerge in antisense frame 1 than in intergenic regions, when the purifying selection is absent, weak and strong, respectively. Interestingly, this analysis revealed that, although asORFs are less likely to emerge in frame 2 than in frame 1, they can emerge more frequently than igORFs. Specifically when the purifying selection is absent, weak and strong, ORFs that are more likely to emerge in antisense frame 2 than in intergenic regions, contain at least 10, 43 and 82 codons, respectively.

However, our analysis of ORF gain probabilities with DNA oligomers estimated from *D. melanogaster* showed that frame 0 has the highest probability of asORF gain (Figure S4B). This finding is qualitatively in agreement with the corresponding probabilities of finding the different asORFs (Fig. 1C).

Purifying selection reduces the number of tolerated mutations in a DNA locus. We note again that even the lowest intensity of purifying selection according to our definition, disallows nonsense mutations

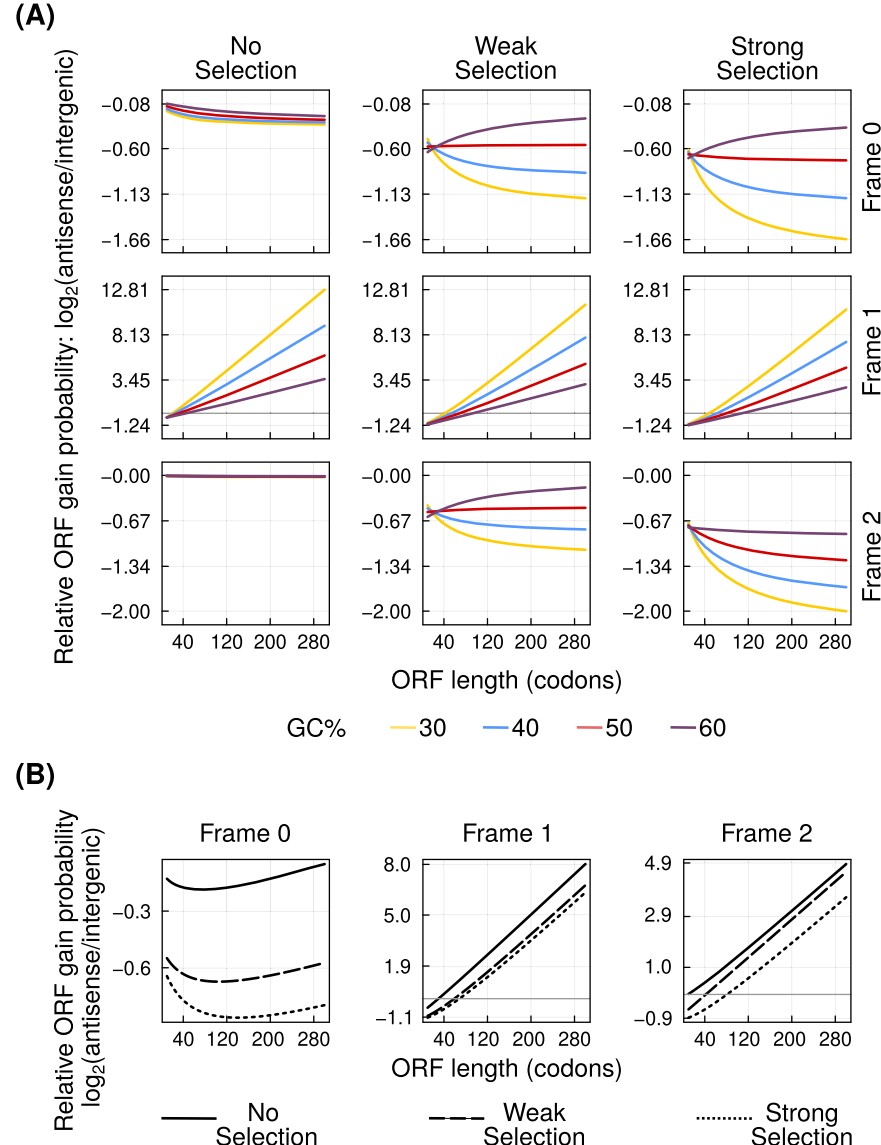

**Fig. 3 | Antisense overlap can facilitate ORF emergence. A** The probability of ORF emergence in the three antisense frames (left to right) relative to that in intergenic regions (log$_2$ ratio, vertical axis), at different intensities of purifying selection (top to bottom). Line colours indicate the GC content of the ORFs. **B** ORFs gain probability in the three antisense frames relative to that in intergenic regions (log$_2$ ratio, vertical axis), calculated using frequencies of short DNA sequences from the yeast genome. Solid, dashed and dotted lines denote zero, weak and strong purifying selection, respectively. Horizontal axis in every plot shows the length of the ORFs. In every plot, we only show asORFs that overlap completely with the sense ORF. In plots where the log ratio spans both positive and negative values, we have highlighted the log ratio of zero using a grey horizontal gridline. Source data are provided as a Source data file.

from occurring in the sense ORFs. We thus hypothesized that overlap with a sense ORF may protect the asORFs from being lost. To this end, we calculated ORF loss probabilities for different ORF lengths, GC content, and intensities of purifying selection (Fig. 4A). In an analogous analysis, we used codon, dicodon, and intergenic trimer frequencies, instead of GC content, to calculate ORF loss probabilities (Fig. 4B). Our analyses show that asORFs are indeed protected from loss due to overlap with existing ORFs, especially when they exist in frame 1. This protection against loss increases with increasing intensity of purifying selection. Our analysis with parameters based on *D. melanogaster* was also in agreement with this result (Figure S5).

To test some of our model's predictions, we analysed the genome and the transcriptome data from the seven different lines of *D. melanogaster*. Six of these lines were obtained from different locations in Europe, whereas one line, the outgroup, was obtained from Zambia[33]. This data set allowed us to analyse gain and loss of transcripts and ORFs in short evolutionary timescales (Supplementary section 6.2, Figure S6).

If an asORF is found in at least one line, it is gained once in *D. melanogaster*. More specifically, the most recently emerged asORF would be detected in only one line, given the assumption that it is not independently lost in six other lines. We found that regardless of whether an asORF is present in one or many lines, they are more abundant in frame 1 than in the other two frames (Fig. 5A). This qualitatively corroborates our model's prediction (especially GC content-based calculation) that antisense overlap in frame 1 facilitates ORF gain (Fig. 3A, Figure S4A).

Next, we analysed the rate of ORF loss in the *D. melanogaster* lines. The genetic variance ($F_{ST}$) between the European populations of *D. melanogaster* is low[34], suggesting that they are not significantly isolated[35]. As a consequence, we could not establish a clear phylogeny for them. Thus, we used a very stringent identification of ORF loss. Specifically, if an ORF is present in the outgroup line (Zambian) and at least one European line, we assume that it was lost in the rest of the European lines. For this definition, we assumed that it is unlikely for an ORF to be gained multiple times independently, and that an ORF can

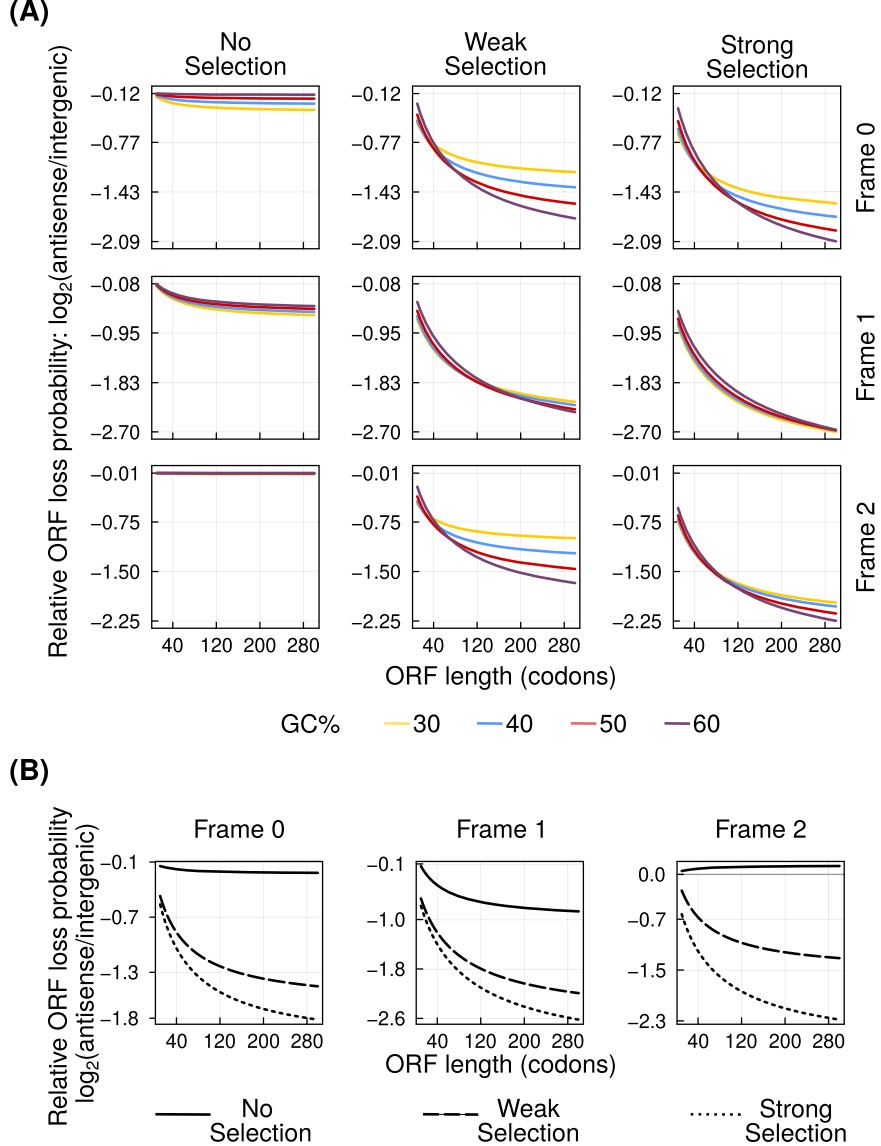

**Fig. 4 | Antisense overlap can reduce ORF loss. A** The probability of ORF loss in the three antisense frames (left to right) relative to that in intergenic regions (log$_2$ ratio, vertical axis), at different intensities of purifying selection (top to bottom). Line colours indicate the GC content of the ORFs. **B** The ORFs loss probability in the three antisense frames relative to that in intergenic regions (log$_2$ ratio, vertical axis), calculated using frequencies of short DNA sequences from the yeast genome. Solid, dashed and dotted lines denote zero, weak and strong purifying selection, respectively. Horizontal axis in every plot shows the length of the ORFs. In every plot, we only show asORFs that overlap completely with the sense ORF. Source data are provided as a Source data file.

be shared between a European line and the outgroup only if it was already present in their common ancestor. To understand the rate of ORF loss, we normalized the number of asORFs lost in any one frame with total number of asORFs present in the same frame. We found that the rate of ORF loss was highest in frame 0, followed by frames 1 and 2 respectively (Fig. 5B). However, the magnitude of this difference was small (<5%) as qualitatively predicted by our model (Fig. 4, Figure S5).

Although antisense overlap can protect ORFs from being lost, it can also constrain the evolution of their sequence. Furthermore, effect of mutations in the sense ORF can also affect different asORFs in the three frames differently. We found that when a sense ORF is under purifying selection (weak or strong), mutational effects are the strongest for asORFs located in frame 2, and the weakest for those in frame 0 (Figure S7).

Overall, our analyses suggest that antisense overlap with an existing ORF facilitates emergence of new ORFs, and protects the existing asORFs from being lost.

## Discussion
To express a protein, a DNA sequence needs to be transcribed as well as translated. New protein coding genes can emerge de novo in non-genic sequences when both these requirements are met. Genomic regions that are already transcribed are thus more likely to evolve protein coding features[19]. Non-coding RNAs indeed harbour ORFs, and some of these ORFs are also actively transcribed, albeit less efficiently than canonical ORFs present in mRNAs[20–22,24]. Several long non-coding RNA genes overlap with other genes in an antisense orientation[29]. This overlap can cause the evolution of asORFs to be constrained by the evolutionary pressures on the corresponding sense genes. The effect of ORF overlap is particularly important in viruses where novel genes frequently emerge overlapping with existing genes, in order to keep the genome compact[30]. In this study, we investigate how likely it is for asORFs to exist in the three possible antisense frames, and how their evolution is constrained by the purifying selection on the sense ORFs. To answer these questions, we developed a mathematical model based

**(A)**

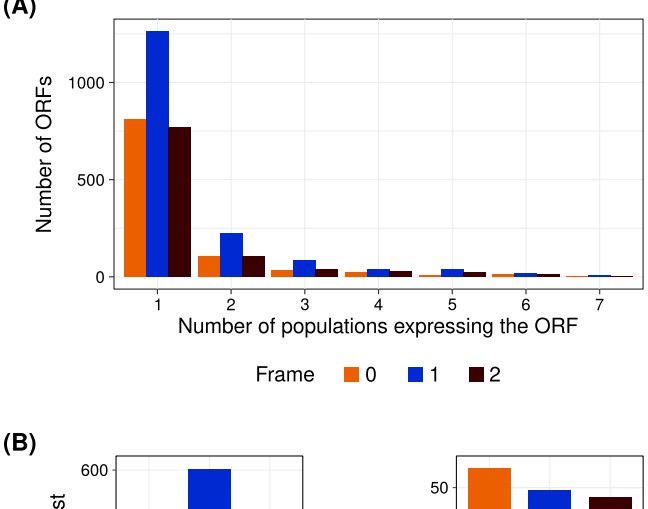

**(B)**

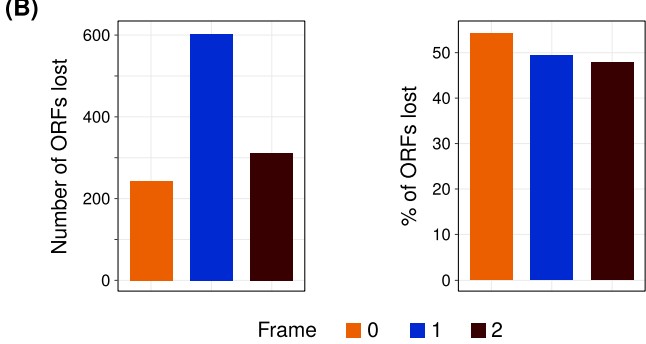

**Fig. 5 | Gain and loss of antisense ORFs in *D. melanogaster* populations. A** Most recently gained asORFs in *D. melanogaster* are frequently located in frame 1. Horizontal axis denotes the number of *D. melanogaster* lines that contain an asORF in their transcriptome (express), and vertical axis denotes the number of such ORFs. **B** *D. melanogaster* asORFs in frame 0 have a higher rate of loss. First panel shows total number of lost ORFs (vertical axis) whereas the second panel shows the percentage of total asORFs that are lost. In all figure panels, the three frames are denoted by three different colours (0: orange, 1: blue, and 2: brown). Source data are provided as a Source data file.

on mutation probabilities, and analysed the genome sequence for validating some of the model's predictions.

Using the model, we show that asORF are more likely to be found in frame 1 than in the other two frames. This prediction is to a large extent supported by our analysis of asORFs in *Saccharomyces cerevisiae* and *Drosophila melanogaster* genomes. Furthermore, asORFs in frame 1 are not only more likely to emerge, but may also be less likely to be lost than asORFs those in the other two frames. More interestingly, ORFs are generally more likely to emerge and to be found in antisense frame 1 than in intergenic regions. Conversely, these asORFs are less likely to be lost than igORFs, due to random mutations. This happens because presence of a sense ORF reduces the chances of premature stop codons occurring in the antisense frame 1.

A previous study has also investigated the effect of selection pressure on different frames, using information theory[31]. Although this study also investigates antisense frames, its analytical approach is different from that of our model. Specifically, we calculate the probability of different kinds of mutations, and focus on the presence or absence of ORFs of different lengths, instead of measuring the fidelity of evolutionary information transfer based on relative rates of synonymous and non-synonymous mutations. Despite these differences in the analytical approach, the findings of our study are in agreement with the previous study. That is, selection pressure on sense ORF causes preservation of asORFs in frame 1[31].

By limiting the number of tolerated mutations, an overlap with an existing ORF can affect the evolution of the protein sequence encoded in an asORF. We quantified mutational effects by estimating the average chemical difference between an original amino acid and a substituted amino acid that results due to random mutations. We found that mutational effects were the strongest in the asORFs in frame 2 (Figure S6). This means that the mutations tolerated in the sense ORFs under purifying selection produce extreme non-synonymous changes in the asORFs in frame 2.

Like all computational models, our model is based on some assumptions and simplifications that need to be considered. For example, we use GC content as a measure of nucleotide composition, which we use in turn to calculate different probability values. For these calculations, we also use codon, dicodon and DNA trimer frequencies, which are data-based measures of nucleotide composition. Our results show that probability values calculated using GC content can sometimes noticeably differ from the values calculated using DNA oligomer distributions, especially for *D. melanogaster*. For example, our estimated probability of finding a *D. melanogaster* asORF was highest in frame 1 when we used GC content, whereas it was highest in frame 0 when we used oligomer distributions. Both our measures of nucleotide composition can vary significantly across the genome (with oligomer frequencies showing more variation; Supplementary Section 8, Figure S8). We used different values of GC content for our calculations that can represent different genomic loci. In contrast, our DNA oligomer-based calculations is based on the average frequency of oligomers from the whole genome. Thus they may not accurately represent any one specific locus. However, our computational framework can be adapted to analyse specific loci. Therefore, model predictions may not be 100% accurate. However, despite the possible inaccuracies, our models are able to produce results that qualitatively agree with real data. Our analyses of asORFs from *S. cerevisiae* and *D. melanogaster* support our model based finding that antisense frame 1 has higher likelihood to harbour asORFs. Our models are based on the assumptions of uniform mutation rate and independence of mutational events. These assumptions are not exactly accurate because mutation rates can vary across the genome[36], and multiple nucleotides can be mutated in a single mutational event[37]. Furthermore, mutation rate bias can be different in different organisms[38,39] (also compare Table 1 and Table S1). Our results show that despite the differences in the mutation rate and mutation rate bias, between yeast and *D. melanogaster*, the results qualitatively remain the same. Thus our predictions are robust to small changes in parameters. It is important to note that our model represents a null hypothesis. It is based on some basic assumptions, and elucidates certain fundamental properties of the genome. However, our data indeed deviate from predictions and sometimes even qualitatively (*D. melanogaster*). Some deviations can be minimized by refining the model with more parameters. However, the reasons for deviation are endless as with any alternative hypothesis. One possible source of deviation could be evolutionary selection, which could cause elimination of some asORFs. Our model can thus can be used to identify the cases where the null hypothesis does not hold true, which can be further studied to test different alternative hypotheses.

We believe our work opens up interesting questions and avenues for future research. For example, the cellular functions and biochemical properties of proteins encoded by asORFs would be worth investigating. This may be especially relevant for antisense lncRNAs, some of which are involved in regulation of gene expression. asORFs may possibly provide another dimension to the cellular function of these RNAs. Translation of ORFs in lncRNAs can indeed be spatiotemporally regulated[22]. asORFs may especially be relevant in organisms with compact genomes, such as viruses. Existing work indeed shows that new protein coding genes emerge in viruses, overlapping with existing genes[30,40,41]. This overlap couples the evolution of the two overlapping genes. Eventually, understanding viral evolution may help design better therapeutic strategies against viral diseases.

**Table 3 | Description of the probability terms used in Equations (1)–(3)**

| Term | Description |
|------|-------------|
| $P_{stop}$ | Probability of finding a stop codon |
| $P_{stop-gain}$ | Probability of gaining a stop codon |
| $P_{stop-loss}$ | Probability of losing a stop codon given that it already exists |
| $P_{stop-stay}$ | Probability that a stop codon exists and is not lost due to mutations |

Here we describe the probabilities associated with stop codons. Analogous probability terms for a start codon are denoted by the subscript, *ATG* (instead of *stop*). For asORFs, $P_{stop}$, $P_{stop-gain}$, $P_{stop-loss}$ and $P_{stop-stay}$ will vary depending on the frame.

## Methods

### Probabilities of finding, gaining, and losing an ORF

We calculated the probabilities of finding, gaining and losing a ORF, using nucleotide composition, mutation rate and mutation rate bias, as described in our previous study[19]. Briefly, a reading frame is an ORF ($P_{ORF}$) when a start codon exists at its beginning ($P_{ATG}$), a stop codon exists at its end ($P_{stop}$), and no stop codon exists in the middle ($1 - P_{stop}$). An ORF emerges ($P_{ORF-gain}$) when two of the three required features are present and are not lost due to mutations, while the missing feature emerges due to mutations. Conversely, an ORF is lost ($P_{ORF-loss}$) when any one of the three required features is lost. The probabilities of finding, gaining and losing an ORF containing $k$ codons, are described by the following equations (Equations (1)–(3)). Table 3 describes the terms used in these equations.

$$P_{ORF}(k) = P_{ATG} \times P_{stop} \times (1 - P_{stop})^{k-2} \tag{1}$$

$$\begin{aligned}
P_{ORF-gain}(k) = {} & P_{ATG-gain} \times P_{stop-stay} \times (1 - P_{stop} - P_{stop-gain})^{k-2} \\
& + P_{ATG-stay} \times P_{stop-gain} \times (1 - P_{stop} - P_{stop-gain})^{k-2} \\
& + P_{ATG-stay} \times P_{stop-stay} \times P_{stop-loss} \times (k-2) \times (1 - P_{stop} - P_{stop-gain})^{k-3}
\end{aligned} \tag{2}$$

$$P_{ORF-loss}(k) = P_{ATG-loss} + P_{stop-loss} + (k-2) \times \frac{P_{stop-gain}}{1 - P_{stop}} \tag{3}$$

### Modelling weak purifying selection

Both gain and loss probabilities of asORFs depend on the strength of selection on the sense ORF. That is, selection would limit the number of sense codons or dicodons that any of the existing codons and dicodons can mutate to. Under strong purifying selection only synonymous mutations are allowed, whereas weak purifying selection allows an amino acid to be substituted by a chemically similar amino acid. To determine chemically similar amino acids, we used an amino acid similarity matrix based on binding covariance of different short peptides to MHC (Major Histocompatibility Complex)[42]. As noted in the original study[42], we identified chemically similar amino acids from pairs of amino acids whose covariance scores are more than 0.05 (Table 4).

### Estimating trimer, codon, and dicodon frequencies

To identify the frequency of all intergenic trimers, we counted all possible trimers in a contiguous stretch of intergenic DNA. Specifically, we used a sliding window approach such that we count trimers starting at every possible position in the DNA sequence. To obtain the frequency of a trimer, we divide the count of each trimer by the total count all trimers. We used the Saccharomyces Genome Database[43], and FlyBase (release 6.4.9)[44], to obtain intergenic regions of *S. cerevisiae* and *D. melanogaster*, respectively.

**Table 4 | Chemically similar amino acids identified using the data from ref. 42**

| Amino acid | Chemically similar amino acids |
|------------|-------------------------------|
| A | P, T, V |
| C | – |
| D | E |
| E | D |
| F | I, W, Y |
| G | – |
| H | K, R |
| I | F, L, M, V |
| K | H, R |
| L | I, M |
| M | I, L |
| N | – |
| P | A |
| Q | – |
| R | H, K |
| S | T |
| T | A, S |
| V | A, I |
| W | F, Y |
| Y | F, W |

To obtain codon and dicodon frequencies, we used a list of non-redundant known ORF sequences of *S. cerevisiae* (SGD)[43] and *D. melanogaster* (FlyBase)[44]. To this end, we counted the nonoverlapping codons (every third position) and dicodons (every sixth position) from the first position of the ORFs.

### Identification of asORFs in the genome

To identify asORFs in *Saccharomyces cerevisiae* genome, we first compiled a list of known antisense RNAs from the S288C reference genome[43], and combined it with the list of novel RNAs identified in a recent study[23]. Next, we identified all ORFs in the combined set of RNAs using the programme *getorf*[32]. Specifically, we identified the longest sequence that starts with the canonical `ATG` start codon and ends with a stop codon. We used a minimum ORF length of 30 nt (default value in *getorf*). We then mapped the genomic coordinates of all the identified ORFs, verified if they overlap with a known ORF in the opposite strand, and calculated the frame of antisense overlap. We used *awk* scripts for this analysis. To calculate the number of ORFs expected from the model, we first identified genomic regions where an antisense overlap exists between an annotated ORF and a RNA. For each such region $A$, with a length $l_A$, we calculated the number of loci (nLoci) where any asORF containing $k$ codons could exist:

$$nLoci(A, k) = \frac{l_A - 3k + 1}{3} \tag{4}$$

$$nLoci(total) = \sum_A \sum_{\substack{k \geq 10 \\ 3k < l_A}} nLoci(A, k) \tag{5}$$

Total number of asORFs in any frame ($f$) would be defined as:

$$N_{asORF}(f) = \sum_A \sum_{\substack{k \geq 10 \\ 3k < l_A}} P_{ORF}(f, k) \, nLoci(A, k) \tag{6}$$

Where $P_{ORF}(f, k)$ is the probability of finding an ORF in a frame $f$ (Fig. 1).

We also identified igORFs from annotated *S. cerevisiae* intergenic regions ($I$)[43] using *getorf*[32]. We calculated the number of intergenic loci where an igORF could exist, and the total number of predicted igORFs as described by the following equations:

$$nLoci(I, k) = l_I - 3k + 1 \qquad (7)$$

$$N_{igORF} = \sum_A \sum_{\substack{k \geq 10 \\ 3k < l_I}} P_{ORF}(k)\, nLoci(I, k) \qquad (8)$$

We performed an analogous analysis for *D. melanogaster*. For details please see Supplementary Section 4.

### Reporting summary

Further information on research design is available in the Nature Portfolio Reporting Summary linked to this article.

## Data availability

*S. cerevisiae* genome sequence used in our study is publicly available on NCBI (release R64, RefSeq accession GCF_000146045.2). We annotated antisense RNAs from a dataset of novel transcripts identified a previous study[23]. The genomic coordinates for these novel transcripts are publicly available on figshare (https://doi.org/10.6084/m9.figshare.7851521.v2). We also include some original data files in the public GitHub repository BharatRaviIyengar/DeNovoEvolution (https://doi.org/10.5281/zenodo.11550958). These files contain the novel transcript sequences (fasta), and their genome positions (gff) as identified in the original study[23]. The genomes and gene annotations of the seven *D. melanogaster* populations that we analysed in this study[33], are publicly available on Zenodo (https://doi.org/10.5281/zenodo.7322757) and on the NCBI BioProject database (PRJNA929424). Source data are provided with this paper.

## Code availability

We implemented our model using Julia programming language and performed data analysis using awk and python programming languages. We provide a brief description of the different programming scripts in Supplementary section 9. All the scripts are publicly available on GitHub: BharatRaviIyengar/DeNovoEvolution (https://doi.org/10.5281/zenodo.11550958).

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

## Acknowledgements

A.G. acknowledges funding from the Alexander von Humboldt Foundation, Deutsche Forschungsgemeinschaft grant BO 2544/20-1 to E.B.-B., and Human Frontier Science Program grant RGP004/2023 to Christine Brun.

## Author contributions

B.R.I. conceived the idea, performed mathematical modelling and statistical tests, analysed the *S. cerevisiae data*, and wrote the manuscript. A.G. performed the analysis of the *D. melanogaster* data, and E.B.-B. procured the funding. All the authors participated in the development of the original idea and in the revision of the manuscript to its current form.

## Funding

## Competing interests

The authors declare no competing interests.
