## [Peer Review File · Nature Communications]

How antisense transcripts can evolve to encode novel proteinsREVIEWER COMMENTS

Reviewer #1 (Remarks to the Author):

This manuscript discussed about an interesting topic of how antisense strand DNAs of coding genes would evolve a de novo gene, a possibility which was raised 10 years ago but did not receive much enthusiasm. This paper reports a systematic analysis and presents an excess of antisense ORFs in frame 1 in both prediction and observation. This conclusion is valuable for further exploration of the de novo gene origination from antisense strand of coding genes. The mathematical model is transparent and easy to test. However, it needs an extensive revision before acceptance.

First, the mathematical model itself needs a test. For example, the model is close to an exponential distribution (equation 1). How well is it fitting to the actual distribution of identified antisense ORFs? If fitting well, the statistical property of exponential distribution can be further used to calculate and explore the properties of the asRNAs.

Second, the reported mutation bias probabilities (Table 1) have a surprisingly high transversion value (G:C  T:A), different from the cited reference (Zhu et al, 2014) and usual understanding of substitution distribution. This is an issue of molecular evolution and has to be confirmed and clarified. The textbook conclusion is a low ratio of transversion over transition.

Third, certain interesting hypotheses were raised but the literature sources were not given. For example, for the antisense overlap lncRNA genes, a more original reference should be referenced: Wu X and Sharp PA, 2013. Cell 155(5): 990-996.

Fourth, the treatment of weak selection is difficult to understand: it is known that synonymous mutation itself is subject to a level of selection and the chemically similar amino acid substitutions are at best neutral.

Reviewer #2 (Remarks to the Author):

In the present manuscript, Iyengar et al. perform an analysis on overlapping antisense ORFs, mostly relying on mathematical modelling. They focus on the constraints that an existing gene imposes on ORF formation and preservation in its antisense strand. The authors place this work in the context of trying to

understand evolutionary processes that give rise to new genes. The most important finding, according to the authors themselves, is that "antisense overlap can increase the likelihood of ORF emergence and reduce the likelihood of ORF loss, especially in one of the three reading frames". I find the motivation behind the manuscript solid, and the general question a very interesting one. I also believe that the mathematical modelling approach proposed here has high potential. But this manuscript falls short of achieving its own goals, and even when it seems to do so things are fuzzier than they should be. I detail below my three major issues:

1) What is the evidence for the main conclusion as stated in the abstract and in the introduction? The first relevant result is described in lines 140-144:

"We found that frame 1 has the highest likelihood of harboring an asORF in most cases..." . But this finding does not reference a graph, so I cannot assess if it is true.

The next relevant part is in lines 149-155. The authors state that:

"We found that the probability of asORFs in frames 0 and 2 were identical to that of the corresponding intergenic ORFs, which means that asORFs in frame 1 are more likely to exist than intergenic ORFs except in the few cases where the ORFs have a GC-content of 60% and less than 39 codons (Figure 1B)"

But Figure 1B does not show any line for frames 0 and 2. Even if the probabilities are purely identical across the entire ORF length range, the lines must be shown. But most importantly, how can the probabilities for frame 2 be identical (not simply similar, mind you) to intergenic regions? How can they be entirely different to those of frame 1, if in both frames 1 and 2 there is overlap with two sense codons and so should be under similar constraint, at least at some level? I find this result puzzling and as far as I can see the authors offer no explanation.

Moving on, the very next paragraph shows results using an alternative way of calculating the probabilities, based on codon and dimer frequencies. A first problem is that there are no methodological details. For example, which genomic parts exactly were considered intergenic? How were they processed to calculate frequencies? Putting that aside however, what is more important is that calculating the probabilities in this way results in huge differences: frame 2 now looks much more like frame 1, and frame 0 becomes less likely to form ORFs than intergenic sequences. What explains such a big difference? Unfortunately I could not find any suggestion by the authors. The results based on GC and dicodons cannot be realistic at the same time.

There's also the additional problem of mismatch between what is shown in Figure 1C and what is written in the text: " we found that asORFs longer than 54 and 75 codons in frames 1 and 2, respectively, are more likely to exist than intergenic ORF of the same lengths (Figure 1C)." but in the figure the lines of frame 1 and 2 are above 0 for the entire range. Same disagreement between text and figure for frame 0.

I hope I haven't misunderstood anything, although it's not impossible because of the lack of methodological detail.

My comments so far relate to the yeast analyses which the authors have chosen to show in the main text. But then there's the comparison with the results in fly. Again, the results based on dicodons are very different, but this time there is an additional difference when comparing yeast and fly: frame 0 goes from equally probable to intergenic in yeast (if we believe figure 1C and not the text), to being the most probable of all three frames. The explanation offered by the authors is that there is biased codon usage in *D. melanogaster* genes. But there is biased codon usage in yeast too, so why the difference between the two species? More generally, I see no reason why equally important results should be relegated to supplementary data, when there is more than enough figure space in the manuscript. The results from fly and yeast should be presented side by side so that we can compare and contrast.

2) My second major point has to do with corroboration using real biological data. Firstly, in Table 2 the expected ORFs for frame 0 are higher than frame 2. I may be mistaken but isn't this the opposite of what the authors say their model predicts? This is an important point and a more detailed comparison here would have been informative. The authors find significant differences between the predictions of their model and the observed number of ORFs. They go on to offer two explanations. If the authors believe that these issues are responsible for the deviation of their model then they should implement them to make it more realistic. Additionally, I fail to see why the authors have chosen to present this analysis using a particularly low length cut-off of 30nt, when their own analyses have shown that the distinction between frames becomes clearer at longer ORF lengths, which are also more likely to produce biologically meaningful products.

It is also unclear to me how the number of loci in Table 2 is calculated and therefore I cannot assess if the observed frequency of intergenic ORFs is indeed lower than that of antisense ones. The frequency itself should also appear in the table.

I found the same issues in the fly comparison and, again, the results should have been presented in the main text. Here, an important deviation from the model is explained once more by "codon usage bias". If the authors have in mind something more specific as to how codon usage bias accounts for longer ORF lengths in frame 0, then they should test it.

Another deviation, and in a sense more worrying, is the fact that intergenic ORFs are more frequent than antisense ones, the opposite of what was found in yeast. The explanation the authors offer is that this could be due to transposable elements and pseudogenes. The authors stop there, but really understanding this difference between model and reality is essential to support the conclusions of this manuscript. Testing this is straightforward and the data are available so I see no reason why not to test it.

The described methodology allowing to corroborate the predictions of the model for gain and loss of ORFs using data from fly is problematic. I'm going to assume that all ORFs >30nt were identified, as for

yeast, even though this is not specified as it should be. The steps to construct orthogroups and infer gain and loss of ORFs are poorly detailed. For example, the authors state "To identify orthogroups, we used nucleotide BLAST(Altschul et al., 1990; Camacho et al., 2009) and restricted alignments to those with a high score and 100% coverage."

Firstly, how high is high score? (bitscore or evalue?) Secondly, BLASTn has known important limitations when applied to short sequences, even at very high similarity so it's not appropriate for this kind of analysis. At the very least the authors should search using protein sequences.

3) A more general comment has to do with the direct implications of the main conclusions of this work. If antisense ORFs in specific frames are more likely to be gained and less likely to be lost than intergenic ones, then it follows that they should be more likely to acquire additional protein-coding features such as translation, and perhaps more likely to acquire function and hence get selected. If the authors disagree with this I'm open to arguments. In my opinion at least, this is an extremely consequential prediction of this model that can and needs to be tested. Catalogs of translated ORFs and ORFs under selection exist for both yeast and fly (e.g. from the Blevins et al. study that the authors already use). Do the data agree with what we might expect if the model was to be correct? If not, why?

General summary of the revisions

Both the reviewers have raised important concerns and have given valuable feedback, that allowed us to significantly improve our manuscript. We thank them both for their valuable suggestions. Before, we proceed to our point by point reply to their comments, we provide a summary of the major changes in the revised version of our manuscript.

- We have addressed all the reviewers' comments by editing the manuscript (main text or supplementary information). We have highlighted the changes in main text using purple font color.
- We have also expanded the supplementary information with four additional sections, two tables and three figures, as suggested by the reviewers.
- We have performed additional statistical analyses to compare the numbers and lengths of antisense ORFs.
- We have removed a para in the introduction, about promoters and poly-A signals because it does not fit with the general theme of the manuscript. It also allowed us to provide space for the additional analyses/explanations that we have now provided in response to reviewers' concerns.
- We have included figure data in our revised submission as supplementary files (in XLSX format).

We hope that our revisions adequately address all the concerns, and is now acceptable for publication.

Reviewer 1

[C1] This manuscript discussed about an interesting topic of how antisense strand DNAs of coding genes would evolve a *de novo* gene, a possibility which was raised 10 years ago but did not receive much enthusiasm. This paper reports a systematic analysis and presents an excess of antisense ORFs in frame 1 in both prediction and observation. This conclusion is valuable for further exploration of the *de novo* gene origination from antisense strand of coding genes. The mathematical model is transparent and easy to test. However, it needs an extensive revision before acceptance.

We thank the reviewer for their careful analysis of our manuscript and relevant comments that allowed us to improve our manuscript.

[C2] First, the mathematical model itself needs a test. For example, the model is close to an exponential distribution (equation 1). How well is it fitting to the actual distribution of identified antisense ORFs? If fitting well, the statistical property of exponential distribution can be further used to calculate and explore the properties of the asRNAs.

The reviewer is correct that the ORF probability decreases exponentially with ORF length. It can be approximated as a geometric distribution which is the discrete analog of the exponential distribution. To visualize the distribution of ORFs, we first calculated the expected number of ORFs for every locus (antisense or intergenic) based on its local GC content. Next, we summed the total number of ORFs in different length bins (interval of 10 codons, starting from 10 codons). We similarly binned the predicted ORFs (getorf). We plotted the expected and observed distribution of antisense ORFs (asORFs, Figure R1A) as well as that of intergenic ORFs (igORFs, Figure R1B), in the different length bins. Both the expected and the observed ORF length distributions, for both asORFs and igORFs appear to be geometrically distributed.

Figure R1: Expected and observed number (vertical axis) of antisense ORFs (A) and intergenic ORFs (B) in *S.cerevisiae* genome. Horizontal axis shows the length range of ORFs (codons).

Because we have an exact equation for ORF probabilities, we need not fit an approximate geometric or an exponential distribution to the data to analyse them. We used these exact ORF probability distributions to calculate the statistics shown in Table 2. We indeed observe some statistically significant differences between the expected and observed frequency of ORFs. These differences could result due to various reasons. For example, the nucleotide distribution may not be as uniform as assumed by the model (both GC content and trimer frequencies). If the observed of ORFs are less numerous than expected then negative selection could be an explanation. We discuss these possibilities in lines 200 and 371 – 385.

[C3] *Second, the reported mutation bias probabilities (Table 1) have a surprisingly high transversion value (G:C → T:A), different from the cited reference (Zhu et al, 2014) and usual understanding of substitution distribution. This is an issue of molecular evolution and has to be confirmed and clarified. The textbook conclusion is a low ratio of transversion over transition.*

We thank the reviewer for pointing this mistake out. We confirm that this was a typological error in the manuscript but the calculations use the correct values. We have now accordingly corrected Table 1.

[C4] *Third, certain interesting hypotheses were raised but the literature sources were not given. For example, for the antisense overlap lncRNA genes, a more original reference should be referenced: Wu X and Sharp PA, 2013. Cell 155(5): 990-996.*

We have now added a reference to the study mentioned by the reviewer, as well as for a few other original studies on antisense lncRNAs (line 50 – 51).

[C5] *Fourth, the treatment of weak selection is difficult to understand: it is known that synonymous mutation itself is subject to a level of selection and the chemically similar amino acid substitutions are at best neutral.*

We agree with the reviewer that even synonymous mutations can have fitness effects as also shown by several studies (for example, Zwart et al. 2018 and Lebeuf-Taylor et al., 2019). Thus chemically similar amino acid substitutions can also have some fitness effects. We agree with the reviewer that they may be neutral in some cases but one can reasonably assume that their chances of being neutral should be smaller than that of synonymous mutations.

To generalize, non-synonymous mutations would have the highest fitness effects, followed by chemically similar substitutions, followed in turn by synonymous mutations. Thus a strong selection would cause the elimination of non-synonymous mutations more frequently, than that of synonymous mutations. Conversely, our three hypothetical selection regimes – no selection, weak selection, and strong selection, allow all mutations, synonymous and chemically similar mutations, and only synonymous mutations, respectively.

More specifically, the weak selection, as defined, allows moderate changes in the bio-

chemical properties of the protein (through chemically similar substitutions), which should result in small differences in phenotype. These chemically similar amino acids are often substituted with each other, as evident by small mismatch penalties in amino acid substitution matrices such as PAM and BLOSUM (as well as the amino acid similarity matrix from Kim *et al.* 2009, that we used in our study). We note that our weak selection is “weaker” than strong selection that preserves the gross biochemistry of the protein molecule (excluding the possibility of cotranslational folding). We don’t intend to generalize our definitions of weak or strong selection, and apply them to other studies on the fitness effects of mutations. We thus conceptually simplify a continuum into 3 categories for the sake of tractability of modelling. This is now clearly pointed out on line 104.

Reviewer 2

[C1] *In the present manuscript, Iyengar et al. perform an analysis on overlapping antisense ORFs, mostly relying on mathematical modelling. They focus on the constraints that an existing gene imposes on ORF formation and preservation in its antisense strand. The authors place this work in the context of trying to understand evolutionary processes that give rise to new genes. The most important finding, according to the authors themselves, is that “antisense overlap can increase the likelihood of ORF emergence and reduce the likelihood of ORF loss, especially in one of the three reading frames”. I find the motivation behind the manuscript solid, and the general question a very interesting one. I also believe that the mathematical modelling approach proposed here has high potential. But this manuscript falls short of achieving its own goals, and even when it seems to do so things are fuzzier than they should be. I detail below my three major issues:*

We thank the reviewer for their careful analysis of our manuscript and relevant comments that allowed us to significantly improve our manuscript. We hope our revised manuscript addresses all their comments.

[C2] *1) What is the evidence for the main conclusion as stated in the abstract and in the introduction? The first relevant result is described in lines 140-144: “We found that frame 1 has the highest likelihood of harboring an asORF in most cases...” . But this finding does not reference a graph, so I cannot assess if it is true.*

We thank the reviewer for pointing this out. We have now reorganized this section based on many other comments from the reviewer. The main conclusion mentioned by the reviewer can be found in lines 137 – 140 of the revised manuscript along with a reference to Figure 1B which displays these findings:

The probabilities of asORFs in frames 0 and 2 are identical for all lengths and GC-content because the overlap does not affect stop codon probability in these frames. This in turn, means that asORFs in these frames are equally probable as intergenic ORFs (igORFs) with identical length and GC-content. This is not the case for frame 1, where we found that asORFs are more likely to be found than in the other two frames and intergenic regions (Figure 1B). The only exceptions are ORFs shorter than 17, 21, 27 and 39 codons present in a DNA region with a GC-content of 30%, 40%, 50% and 60%, respectively. Even for these exceptional cases, the probability of an asORF in frame 1 is no less than 74% of the corresponding ORF probabilities in the other frames.

[C3] *The next relevant part is in lines 149-155. The authors state that: “We found that the probability of asORFs in frames 0 and 2 were identical to that of the corresponding intergenic ORFs, which means that asORFs in frame 1 are more likely to exist than intergenic ORFs except in the few cases where the ORFs have a GC-content of 60% and less than 39*

codons (Figure 1B)". But Figure 1B does not show any line for frames 0 and 2. Even if the probabilities are purely identical across the entire ORF length range, the lines must be shown.

We indeed tried adding the lines corresponding to frames 0 and 2 as the reviewer suggested. Because the probabilities of antisense and intergenic ORFs are identical for all values of GC content, the different lines are superimposed (giving an illusion that not all data was plotted). Therefore, in our opinion, adding frames 0 and 2 does not make the plot very informative. Therefore, we do not change the figure in this aspect. We have instead provided a detailed textual explanation on why the probabilities of asORFs and igORFs are identical. Please see our next reply.

[C4] But most importantly, how can the probabilities for frame 2 be identical (not simply similar, mind you) to intergenic regions? How can they be entirely different to those of frame 1, if in both frames 1 and 2 there is overlap with two sense codons and so should be under similar constraint, at least at some level? I find this result puzzling and as far as I can see the authors offer no explanation.

Before we address the reviewer's question, we note that the coding and the intergenic regions that are being compared have identical GC content. We have also mentioned this consideration in lines 137 – 140, and Figure 1 captions. We further

(A)

TAA		TAG	
AAT TAA	AAT TAG	AAC TAA	AAC TAG
TAT TAA	TAT TAG	TAC TAA	TAC TAG
GAT TAA	GAT TAG	GAC TAA	GAC TAG
CAT TAA	CAT TAG	CAC TAA	CAC TAG
ATT TAA	ATT TAG	ATC TAA	ATC TAG
TTT TAA	TTT TAG	TTC TAA	TTC TAG
GTT TAA	GTT TAG	GTC TAA	GTC TAG
CTT TAA	CTT TAG	CTC TAA	CTC TAG
AGT TAA	AGT TAG	AGC TAA	AGC TAG
TGT TAA	TGT TAG	TGC TAA	TGC TAG
GGT TAA	GGT TAG	GGC TAA	GGC TAG
CGT TAA	CGT TAG	CGC TAA	CGC TAG
ACT TAA	ACT TAG	ACC TAA	ACC TAG
TCT TAA	TCT TAG	TCC TAA	TCC TAG
GCT TAA	GCT TAG	GCC TAA	GCC TAG
CCT TAA	CCT TAG	CCC TAA	CCC TAG

(B)

TAA				TAG				TGA			
AAT AAA	ATT AAT	ATT AAG	ATT AAC	ACT AAA	ACT AAT	ACT AAG	ACT AAC	ATC AAA	ATC AAT	ATC AAG	ATC AAC
TTT AAA	TTT AAT	TTT AAG	TTT AAC	TCT AAA	TCT AAT	TCT AAG	TCT AAC	TTC AAA	TTC AAT	TTC AAG	TTC AAC
GTT AAA	GTT AAT	GTT AAG	GTT AAC	GCT AAA	GCT AAT	GCT AAG	GCT AAC	GTC AAA	GTC AAT	GTC AAG	GTC AAC
CTT AAA	CTT AAT	CTT AAG	CTT AAC	CCT AAA	CCT AAT	CCT AAG	CCT AAC	CTC AAA	CTC AAT	CTC AAG	CTC AAC
ATT ATA	ATT ATT	ATT ATG	ATT ATC	ACT ATA	ACT ATT	ACT ATG	ACT ATC	ATC ATA	ATC ATT	ATC ATG	ATC ATC
TTT ATA	TTT ATT	TTT ATG	TTT ATC	TCT ATA	TCT ATT	TCT ATG	TCT ATC	TTC ATA	TTC ATT	TTC ATG	TTC ATC
GTT ATA	GTT ATT	GTT ATG	GTT ATC	GCT ATA	GCT ATT	GCT ATG	GCT ATC	GTC ATA	GTC ATT	GTC ATG	GTC ATC
CTT ATA	CTT ATT	CTT ATG	CTT ATC	CCT ATA	CCT ATT	CCT ATG	CCT ATC	CTC ATA	CTC ATT	CTC ATG	CTC ATC
ATT AGA	ATT AGT	ATT AGG	ATT AGC	ACT AGA	ACT AGT	ACT AGG	ACT AGC	ATC AGA	ATC AGT	ATC AGG	ATC AGC
TTT AGA	TTT AGT	TTT AGG	TTT AGC	TCT AGA	TCT AGT	TCT AGG	TCT AGC	TTC AGA	TTC AGT	TTC AGG	TTC AGC
GTT AGA	GTT AGT	GTT AGG	GTT AGC	GCT AGA	GCT AGT	GCT AGG	GCT AGC	GTC AGA	GTC AGT	GTC AGG	GTC AGC
CTT AGA	CTT AGT	CTT AGG	CTT AGC	CCT AGA	CCT AGT	CCT AGG	CCT AGC	CTC AGA	CTC AGT	CTC AGG	CTC AGC
ATT ACA	ATT ACT	ATT ACG	ATT ACC	ACT ACA	ACT ACT	ACT ACG	ACT ACC	ATC ACA	ATC ACT	ATC ACG	ATC ACC
TTT ACA	TTT ACT	TTT ACG	TTT ACC	TCT ACA	TCT ACT	TCT ACG	TCT ACC	TTC ACA	TTC ACT	TTC ACG	TTC ACC
GTT ACA	GTT ACT	GTT ACG	GTT ACC	GCT ACA	GCT ACT	GCT ACG	GCT ACC	GTC ACA	GTC ACT	GTC ACG	GTC ACC
CTT ACA	CTT ACT	CTT ACG	CTT ACC	CCT ACA	CCT ACT	CCT ACG	CCT ACC	CTC ACA	CTC ACT	CTC ACG	CTC ACC

Table R1: (A) The 64 sense dicodons that contain a stop codon, and that overlap with an antisense stop codon in frame-1. (B) The 192 sense dicodons overlapping an antisense stop codon in frame-2. We have highlighted in red font the reverse complementary sequence corresponding to an antisense stop codon.

note that probabilities of ORFs are primarily determined by the probability of stop codons (lines 115 –118).

The probability of finding an antisense stop codon in frame 0 is same as the probability of finding the three reverse complementary codons in the sense ORF (TTA, CTA and TCA). These three (reverse complementary) codons are allowed in the sense ORFs, and their probability would be simply determined by the GC content of the sense ORF. These three codons have the same GC composition as the stop codons, and therefore, their probability is identical to that of stop codons (given identical GC content of the locus). Therefore, given these considerations, the probability of a frame-0 asORF is identical to that of an intergenic ORF of same length and GC content.

Next, we explain why the probability of frame-2 asORFs is identical to that of intergenic ORFs of similar nucleotide composition and length. The probability of finding a frame-2 antisense stop codon is determined by the corresponding dicodons in the sense ORF. There are 64 possible overlapping dicodons for both frame 1 and frame 2 antisense codons ($4^3 = 64$; three out of six positions in a dicodon are determined by the overlapping antisense codon). Thus, there are $64 \times 3 = 192$ dicodons that overlap with any of the three antisense stop codons. By definition, the sense ORF should not contain a stop codon which means that no dicodon can contain a stop codon. For frame-1 antisense stop codons, 64 overlapping sense overlapping dicodons contain a stop codon (Table R1A), whereas for frame-2 antisense stop codons none of the overlapping dicodons contain a stop codon (Table R1B). Therefore, the probability of an antisense frame-2 stop codon is identical to that of a stop codon in an intergenic locus with identical GC content.

We now briefly explain in the main text (lines 121 – 131) why probability of stop codons in antisense frames 0 and 2 is unaffected by the overlap. We provide a more detailed explanation in the supplement (Section 2; Table R1 is the new Table S2).

[C5] Moving on, the very next paragraph shows results using an alternative way of calculating the probabilities, based on codon and dimer frequencies. A first problem is that there are no methodological details. For example, which genomic parts exactly were considered intergenic? How were they processed to calculate frequencies?

We have now expanded the methods section to explain the procedure used for calculating oligomer frequencies (lines 433 – 440).

[C6] Putting that aside however, what is more important is that calculating the probabilities in this way results in huge differences: frame 2 now looks much more like frame 1, and frame 0 becomes less likely to form ORFs than intergenic sequences. What explains such a big difference? Unfortunately I could not find any suggestion by the authors. The results based on GC and dicodons cannot be realistic at the same time.

The reviewer's question about which parameter sets are more realistic, is very relevant. On a general note, any calculation made using an averaged nucleotide compo-

Figure R2: Variance of the normalized distribution of GC content and of different DNA trimers in *S. cerevisiae*. For coding regions we calculated the frequencies of the different codons as they exist in annotated ORFs (top panel), whereas for regions overlapping with antisense ORFs, we calculated the distribution of DNA trimers using a sliding window (bottom panel). We have excluded stop codons from both the panels.

sition distribution is likely to be an approximation. It is true for both GC content (for example, using the average genomic GC content) or average distribution of DNA oligomers across different genomic loci. Both GC content and oligomer distribution can be calculated for specific loci, which make the analysis more realistic. In our plots for of stationary, gain and loss probability based on GC content (Figures 1B, 3A and 4A), we show four different values of GC content. The goal of these plots is not to show what actually happens in the specific genome but to establish general principles based on some basic assumptions. Hence, they are correct as long as our assumptions hold true. The plots based on DNA oligomer frequencies (Figures 1C, 3B and 4B) may be less realistic because they assume that the oligomer distribution is uniform across the genome (CDS or intergenic regions). Thus the GC content based plots are more informative.

To understand how realistic averages can be, we performed an empirical analysis of variance of nucleotide composition. Specifically, we normalized the distribution such that the sum of frequencies of a trimer (or GC fraction) across all loci is equal to one, and calculated the variance of this distribution. We found that GC content has a smaller variance than that of any DNA trimer (Figure R2). This empirical analysis however does not suggest that GC content is a better estimate of the real nucleotide distribution.

Ultimately, the most realistic analysis would estimate parameters from each locus separately, and estimate the ORF probabilities specific to that locus. We have indeed done so for calculating expected number of ORFs based on GC content (Table 2). To this end, we calculated the GC content of each contiguous intergenic or antisense

overlapping region, and estimated the ORF probability as well as expected number of ORFs using this specific GC content. We found that the expected number of ORF using global DNA trimer distribution and locus specific GC content do not differ significantly. Moreover, the expected number of ORFs, although statistically significantly different from the observed values in some cases, were reasonably close to the latter. Thus we think that our approximations are reasonable for the questions addressed in our study. Please note that Table 2 is now revised because we have used a more detailed analysis as outlined above.

We briefly explain how to choose more realistic parameters, in the discussion (lines 371 – 385).

We have also included a more detailed analysis of variation (as written in this reply) in the supplement (Section 8).

[C7] *There's also the additional problem of mismatch between what is shown in Figure 1C and what is written in the text: "we found that asORFs longer than 54 and 75 codons in frames 1 and 2, respectively, are more likely to exist than intergenic ORF of the same lengths (Figure 1C)." but in the figure the lines of frame 1 and 2 are above 0 for the entire range. Same disagreement between text and figure for frame 0. I hope I haven't misunderstood anything, although it's not impossible because of the lack of methodological detail.*

We thank the reviewer for finding out this mistake, and apologize for the confusing text. We have also realized that we had plotted the probability of ORFs ranging from 30 to 300 codons. We have now increased the ORF length range to 10 – 300 codons, consistent with the minimum ORF size used in our genomic data analysis (Figure 2). We have corrected the text (lines 140 –142) which now reads:

This is not the case for frame 1, where we found that asORFs are more likely to be found than in the other two frames and intergenic regions (Figure 1B). The only exceptions are ORFs shorter than 17, 21, 27 and 39 codons present in a DNA region with a GC-content of 30%, 40%, 50% and 60%, respectively.

...

...

We also calculated the probability of asORFs using actual codon and dicodon frequencies in annotated yeast ORFs. Likewise, we calculated the probability of igORFs using the frequencies of DNA trimers in yeast intergenic genome. With this analysis, we found that asORFs longer than 17, 21, and 19 codons, in frames 0, 1 and 2, respectively, are more likely to exist than igORFs of the same lengths (Figure 1C)

[C8] *My comments so far relate to the yeast analyses which the authors have chosen to show in the main text. But then there's the comparison with the results in fly. Again, the results based on dicodons are very different, but this time there is an additional difference when comparing yeast and fly: frame 0 goes from equally probable to intergenic in yeast (if we believe*

Figure R3: Codon usage of leucine and serine in *S. cerevisiae* and *D. melanogaster*. Codons highlighted in bold overlap with an antisense stop codon.

figure 1C and not the text), to being the most probable of all three frames. The explanation offered by the authors is that there is biased codon usage in *D. melanogaster* genes. But there is biased codon usage in yeast too, so why the difference between the two species?

We thank the reviewer for raising this concern. As we mentioned in our reply to comment C7, we have changed the size range of ORFs for which we compute the probabilities. The revised figures should clarify at least some questions. Specifically, frame 0 becomes the most probable only when the ORFs are long. However, most ORFs are short (ORF probability reduces exponentially with ORF length) for which the differences between the ORF probabilities in the three frames are not so drastic.

Having said that, we proceeded to investigate why frame 0 becomes the best location for asORFs (at least for long ORFs) when we use codon frequencies of *D. melanogaster*. We analysed the differences between the predictions from the two species more closely. As the reviewer noted, the most salient difference exists in the probability of asORFs in frame 0. The reason is that stop codons in frame 0 are 2.7 times more likely in *S. cerevisiae* than in *D. melanogaster* (Table R2). Therefore we analysed the frequency of these codons and their specific usage to encode the corresponding amino acids.

	S. cerevisiae	D. melanogaster
Start codon	0.0169	0.0172
Stop codon: Frame 0	0.0592	0.0216
Stop codon: Frame 1	0.0399	0.0319
Stop codon: Frame 2	0.0482	0.0423

Table R2: Probability of start and stop codons in the three different antisense frames, calculated using distribution of codons and dicodons in *S. cerevisiae* and *D. melanogaster* coding sequences.

Stop codons in frame 0 overlap with the codons – TTA, CTA (coding for leucine) and TCA (coding for serine). Both leucine and serine are encoded by six codons. We analysed the coding regions of *S. cerevisiae* and *D. melanogaster* to estimate the codon usage for leucine and serine in both these organisms. We found that the total frequencies of leucine and serine are similar between the two organisms. However, the codons that overlap with an antisense stop codon are more frequently used in *S. cerevisiae* than in *D. melanogaster* (Figure R3). This biased codon usage for leucine and serine, between the two organisms, causes the difference in the predicted frequency of asORFs in frame 0 using codon frequencies. We have now included a brief summary of the above analysis, in the main text (lines 161 – 166) and provided a more detailed explanation in the supplement (Section 3).

[C9] *More generally, I see no reason why equally important results should be relegated to supplementary data, when there is more than enough figure space in the manuscript. The results from fly and yeast should be presented side by side so that we can compare and contrast.*

We have now expanded figure 1 to include data from both yeast and fly. However, figures 3 and 4 are already huge. We understand the reviewer's suggestion that including data from both organisms will help a reader note the differences between the results from the two organisms. However, it would also make the manuscript bulkier, that may not fit well with the compact format of *Nature Communications*. Furthermore, the main goal of this study is to explain a general model that can be (and often must be) adapted to specific data. Therefore, we believe that too much information in the main text may obscure the basic concept. However, if both the reviewer(s) and the editor(s) agree that all figures should be included in the main text, we would be happy to oblige.

[C10] *2) My second major point has to do with corroboration using real biological data. Firstly, in Table 2 the expected ORFs for frame 0 are higher than frame 2. I may be mistaken but isn't this the opposite of what the authors say their model predicts? This is an important point and a more detailed comparison here would have been informative. The authors find significant differences between the predictions of their model and the observed number of ORFs. They go on to offer two explanations. If the authors believe that these issues are responsible for the deviation of their model then they should implement them to make it more realistic.*

We thank the reviewer for pointing this mistake out. It was a calculation error that we have now corrected. Furthermore, we have now revised the table as described in our replies to comment C8. The predicted and observed values are now in agreement.

[C11] *Additionally, I fail to see why the authors have chosen to present this analysis using a particularly low length cut-off of 30nt, when their own analyses have shown that the distinction between frames becomes clearer at longer ORF lengths, which are also more likely to produce biologically meaningful products.*

We chose a small cutoff because *de novo* protein coding genes are known to be small

(see Van Oss and Carvunis, 2019). Some recent studies have investigated short ORFs (Finkel et al., 2017; Patraquim et al., 2020; Leong et al., 2022) but their biological roles for the majority of short ORFs is yet to be characterized. It is a new field and several open questions may be addressed in near future. Short ORFs are also more likely to emerge in short evolutionary timescale. Therefore we believe that short ORFs are worth investigating.

[C12] *It is also unclear to me how the number of loci in Table 2 is calculated and therefore I cannot assess if the observed frequency of intergenic ORFs is indeed lower than that of antisense ones. The frequency itself should also appear in the table.*

We have revised the method section to explain how the number of loci were calculated (lines 444 – 456, New Equations 4–8). We have also included the frequencies in the new Table 2.

[C13] *I found the same issues in the fly comparison and, again, the results should have been presented in the main text. Here, an important deviation from the model is explained once more by “codon usage bias”. If the authors have in mind something more specific as to how codon usage bias accounts for longer ORF lengths in frame 0, then they should test it.*

Please see our reply to comment C8.

[C14] *Another deviation, and in a sense more worrying, is the fact that intergenic ORFs are more frequent than antisense ones, the opposite of what was found in yeast. The explanation the authors offer is that this could be due to transposable elements and pseudogenes. The authors stop there, but really understanding this difference between model and reality is essential to support the conclusions of this manuscript. Testing this is straightforward and the data are available so I see no reason why not to test it.*

We apologize for this oversight. We now provide a better explanation for why the antisense ORFs that we analysed are less frequent than intergenic ORFs in *D. melanogaster*.

We believe that our stringent analysis severely restricts the search space of antisense ORFs. We restricted our search space such that we discard asORFs that are multi-exonic. The reason is that introns can change the frame of overlap between the flanking exons. Thus any asORF cannot be assigned one frame. We also restrict our analysis to asORFs that completely overlap with a protein coding exon. These stringent criteria do not change the distribution of asORFs in the different frames. We estimated the antisense regions as the set of transcribed exons that antisense-overlap with any protein coding exon. The number of asORFs found are not significantly different from expected. However, the restricted search space for asORFs only allows short ORFs to be detected. Here we would like to remind the reviewer that the probability of short asORFs is less than that of igORFs. This could be the reason why the expected frequency of asORFs is smaller than that of igORFs in *D. melanogaster*. We now provide a brief explanation in the main text (lines 234 – 240).

[C15] *The described methodology allowing to corroborate the predictions of the model for gain and loss of ORFs using data from fly is problematic. I'm going to assume that all ORFs >30nt were identified, as for yeast, even though this is not specified as it should be. The steps to construct orthogroups and infer gain and loss of ORFs are poorly detailed. For example, the authors state "To identify orthogroups, we used nucleotide BLAST(Altschul et al., 1990; Camacho et al., 2009) and restricted alignments to those with a high score and 100% coverage." Firstly, how high is high score? (bitscore or evalue?) Secondly, BLASTn has known important limitations when applied to short sequences, even at very high similarity so it's not appropriate for this kind of analysis. At the very least the authors should search using protein sequences.*

The reviewer has raised many relevant points. We used BLASTN for a specific reason – we wanted to identify orthologous asORFs that may be frameshifted. In case of a frameshift, BLASTP may not detect any homology. We used an e-value cutoff of 10^{-2} and required a 100% query coverage. Furthermore, we verified that the orthologous asORFs are antisense to the same protein coding gene. Given these criteria, our algorithm picks the highest scoring hit if there are multiple hits. To keep the analysis focused and less complicated, we only analysed asORF orthologs in which the frame was conserved. Thus our BLAST analysis is overall quite stringent. We have now revised the methods description in supplementary section 4.

[C16] 3) *A more general comment has to do with the direct implications of the main conclusions of this work. If antisense ORFs in specific frames are more likely to be gained and less likely to be lost than intergenic ones, then it follows that they should be more likely to acquire additional protein-coding features such as translation, and perhaps more likely to acquire function and hence get selected. If the authors disagree with this I'm open to arguments. In my opinion at least, this is an extremely consequential prediction of this model that can and needs to be tested. Catalogs of translated ORFs and ORFs under selection exist for both yeast and fly (e.g. from the Blevins et al. study that the authors already use). Do the data agree with what we might expect if the model was to be correct? If not, why?*

We thank the reviewer for this suggestion. Before we address the question, we would like to point out that evolution of protein coding features depends on several other factors. For example, the properties of the encoded protein can drive selection. One can also rationalize that evolution of a certain feature does not primarily depend on the emergence probability but that of number of sites where it can emerge. Assuming that the probability of evolution of a certain feature is uniform across all loci, then the evolutionary rate should depend on the total number of loci. That is, evolutionary rate of more abundant igORFs should be higher than the less abundant asORFs, even if the probability of asORF (frame 1) at any one site is higher than that of igORF in a site with similar size and composition.

Given these considerations, we performed an analysis of translation efficiency of ORFs. We found that although asORFs are most numerous in frame 1, and can produce more translated products, any random asORFs in frame 1 is not likely to be more efficiently translated than asORFs in frames 0 and 2. Specifically, the median riboseq reads (in *S.cerevisiae*) or Kozak consensus sequence strength of asORFs (*D.*

melanogaster) is not higher in case of frame 1 compared to the other two frames. However, iGORFs are significantly more efficiently translated than asORFs in *S. cerevisiae* (based on riboseq reads). There could be several reasons for this observation. First, an abundance of igORFs can facilitate emergence of “functional” features in at least some of the ORFs, which can lead to their selection (higher rate of evolution). Second, diversification of igORFs is also more feasible because they are not constrained by selection on the overlapping gene. Finally, a large majority of intergenic ORFs could be genuine protein coding genes. Highly expressed protein coding genes that are annotated as intergenic regions could be part of transposable regions.

We did not find any significant difference between the strength of Kozak consensus sequences of asORFs and igORFs of *D. melanogaster*.

We now include a detailed analysis in the supplement (Section 5) and provide a brief summary in the main text (lines 241 – 249). We note that a lack of a significant difference in translational efficiency does not indicate that there is no significant difference in the total translational output. That is so because both the number of asORFs and the translational efficiency is responsible for translational output. We found that the total translational output is significantly higher for asORFs in frame 1 than those in the other two frames (one tailed Fisher exact test, FDR corrected $P < 10^{-22}$).

REVIEWER COMMENTS

Reviewer #1 (Remarks to the Author):

I am satisfied with the revisions, which have clearly addressed the several issues I raised in the previous version. I recommend the current revised version to Nature Comm for publication.

Reviewer #2 (Remarks to the Author):

I am reviewing the revised version after having reviewed the original submission.

I will therefore focus on the revisions and on the replies to my initial comments.

C2) I'm glad the authors fixed this.

C3-C4) I understand the authors' explanation about the co-occurrence of stop codons on the different frames. Looking at how the authors calculate the probability of finding an ORF, this initially seems to be sufficient. Yet an ORF has internal codons as well, not just a start and a stop. And by definition the existence of some di-codons in the +2 antisense frame means existence of stop codons in the sense ORF something which is also true for the three complementary codons of 0 antisense frame (TTA, CTA, TCA). In both cases, this should create a constraint and my expectation is that it is very unlikely that these two constraints will be exactly identical. From what I understand (again there is a paucity of methodological details on this point and things were not made any clearer by searching within the authors' previous publication cited in the methods) this fact is not and in fact cannot be reflected in the model which only works with probabilities of stop and start codons. So if this is the case and I have not missed something here, does this not strongly limit the usefulness of this approach to this specific problem? If the answer is yes, then this needs to be fully articulated in the discussion.

C5) There is still not enough detail in this part of the methods to reproduce this analysis.

C6) I agree that using locus-specific parameters makes things more realistic. But that wasn't my point in this comment. Before comparing to the real data, the authors present the results of their model based on these two different types of parameters. My point was that both these non-locus-specific results

cannot be true at the same time, and so a justification about why is expected. I fail to see this in the authors' response.

C7) I'm glad the authors corrected this.

C8) I appreciate the authors explanation.

C9) Figure 2 however is not huge, and the *D. melanogaster* results for B could easily be included to facilitate the comparison.

C10) I'm glad the authors corrected the initial error. But in their reply the authors state that "predicted and observed values are now in agreement", contradicting their own text: "We found that the actual asORFs in the yeast genome were 1.6 – 24% fewer than expected". At the very least the comparison to the real data shows that the model is limited and this should be very clearly presented in the discussion. (it isn't currently)

C11) I too believe that short ORFs are worth investigating; but that wasn't the comment. Given the model predictions, one would expect a sharper difference between the frames if the analyses were restricted only to longer ORFs. Why not test this directly when all the data are available to do so?

C12) This is OK.

C13) The reply to C8 does not cover any deviation of the model from real data.

C14) So from what I understand this is a totally different explanation from the initial one, but it is still a speculative one. My comment still holds: this is so central to the study's message that it needs to be tested.

C15) I'm glad the authors provided the missing methodological details. TBLASTN can also detect frameshifts and is more sensitive than BLASTN. But in any case, if the idea is to be as stringent as possible I guess it is OK.

C16) I appreciate the additional analyses, which seem to suggest a complicated relationship between probability of an ORF forming and probability of it evolving into a gene.

We thank the reviewer again for a thorough analysis of our manuscript and giving valuable feedback. Here, we present our responses to the pending concerns hoping that we have addressed them satisfactorily.

C3-C4) *I understand the authors' explanation about the co-occurrence of stop codons on the different frames. Looking at how the authors calculate the probability of finding an ORF, this initially seems to be sufficient. Yet an ORF has internal codons as well, not just a start and a stop. And by definition the existence of some di-codons in the +2 antisense frame means existence of stop codons in the sense ORF something which is also true for the three complementary codons of 0 antisense frame (TTA, CTA, TCA). In both cases, this should create a constraint and my expectation is that it is very unlikely that these two constraints will be exactly identical. From what I understand (again there is a paucity of methodological details on this point and things were not made any clearer by searching within the authors' previous publication cited in the methods) this fact is not and in fact cannot be reflected in the model which only works with probabilities of stop and start codons. So if this is the case and I have not missed something here, does this not strongly limit the usefulness of this approach to this specific problem? If the answer is yes, then this needs to be fully articulated in the discussion.*

We apologize that we couldn't be any clearer in our previous response. We'll address this concern step by step, answering each point in the same order as the reviewer raised them. We include the dicodon table from the previous review to explain our answers better.

1. We reiterate that because a sense ORF cannot contain stop codons, it reduces the probability of antisense stop codon in frame 1 (but not in 0 and 2). The reviewer agrees with this logic if we have understood correctly. Their concern pertains to internal codons.

TAA				TAG				TGA			
ATT AAA	ATT AAT	ATT AAG	ATT AAC	ACT AAA	ACT AAT	ACT AAG	ACT AAC	ATC AAA	ATC AAT	ATC AAG	ATC AAC
TTT AAA	TTT AAT	TTT AAG	TTT AAC	TCT AAA	TCT AAT	TCT AAG	TCT AAC	TTC AAA	TTC AAT	TTC AAG	TTC AAC
GTT AAA	GTT AAT	GTT AAG	GTT AAC	GCT AAA	GCT AAT	GCT AAG	GCT AAC	GTC AAA	GTC AAT	GTC AAG	GTC AAC
CCT AAA	CCT AAT	CCT AAG	CCT AAC	CCT AAA	CCT AAT	CCT AAG	CCT AAC	CTC AAA	CTC AAT	CTC AAG	CTC AAC
ATT ATA	ATT ATT	ATT ATG	ATT ATC	ACT ATA	ACT ATT	ACT ATG	ACT ATC	ATC ATA	ATC ATT	ATC ATG	ATC ATC
TTT ATA	TTT ATT	TTT ATG	TTT ATC	TCT ATA	TCT ATT	TCT ATG	TCT ATC	TTC ATA	TTC ATT	TTC ATG	TTC ATC
GTT ATA	GTT ATT	GTT ATG	GTT ATC	GCT ATA	GCT ATT	GCT ATG	GCT ATC	GTC ATA	GTC ATT	GTC ATG	GTC ATC
CCT ATA	CCT ATT	CCT ATG	CCT ATC	CCT ATA	CCT ATT	CCT ATG	CCT ATC	CTC ATA	CTC ATT	CTC ATG	CTC ATC
ATT AGA	ATT AGT	ATT AGG	ATT AGC	ACT AGA	ACT AGT	ACT AGG	ACT AGC	ATC AGA	ATC AGT	ATC AGG	ATC AGC
TTT AGA	TTT AGT	TTT AGG	TTT AGC	TCT AGA	TCT AGT	TCT AGG	TCT AGC	TTC AGA	TTC AGT	TTC AGG	TTC AGC
GTT AGA	GTT AGT	GTT AGG	GTT AGC	GCT AGA	GCT AGT	GCT AGG	GCT AGC	GTC AGA	GTC AGT	GTC AGG	GTC AGC
CCT AGA	CCT AGT	CCT AGG	CCT AGC	CCT AGA	CCT AGT	CCT AGG	CCT AGC	CTC AGA	CTC AGT	CTC AGG	CTC AGC
ATT ACA	ATT ACT	ATT ACC	ATT ACC	ACT ACA	ACT ACT	ACT ACC	ACT ACC	ATC ACA	ATC ACT	ATC ACC	ATC ACC
TTT ACA	TTT ACT	TTT ACC	TTT ACC	TCT ACA	TCT ACT	TCT ACC	TCT ACC	TTC ACA	TTC ACT	TTC ACC	TTC ACC
GTT ACA	GTT ACT	GTT ACC	GTT ACC	GCT ACA	GCT ACT	GCT ACC	GCT ACC	GTC ACA	GTC ACT	GTC ACC	GTC ACC
CCT ACA	CCT ACT	CCT ACC	CCT ACC	CCT ACA	CCT ACT	CCT ACC	CCT ACC	CTC ACA	CTC ACT	CTC ACC	CTC ACC

Table R1: The 192 sense dicodons overlapping an antisense stop codon in frame-2. We have highlighted in red font the reverse complementary sequence corresponding to an antisense stop codon.

2. Yes we do take into account the internal codons for the calculation, but not directly. The internal codons are expected at frequencies based on the nucleotide distribution of the locus. We first model their nucleotide distribution using GC content (also for dicodons). Now if any codon but a stop codon can exist in the sense ORF, and if the codons are distributed according to GC content alone (please note that we do not talk about specific ORFs), then the frequency of TTA, CTA and TCA (anti-stop 0), would be $W^3 + 2SW^2$ (where $W = A$ or T , and $S = G$ or C). In Table R1 that shows all dicodons overlapping with a stop codon in the antisense frame 2 (red sequence). Here the red sequence is flanked by all possible nucleotide/dinucleotides: 3 (antistop) $\times 4^3$ (flanks) = 192. The sum of all possibilities in the flank would be equal to 1, and hence the probability of anti-stop 2 is same as that of TTA, CTA and TCA (red part in Table R1) = $W^3 + 2SW^2$. So the probability of asORFs in frames 0 and 2 are **exactly identical** if one models nucleotide distribution using GC content. The necessary methodology is simply the above calculation. It is also explained in our previous study (second section of methods).

3. While mathematically the probability of asORF0 and asORF2 are identical (for a given value of GC content), it sounds unlikely the two probabilities should be identical as the reviewer notes. It certainly depends on the composition of an ORF. Therefore, we calculated an average codon and dicodon distribution from a list of non-redundant ORF sequences from *S. cerevisiae* and *D. melanogaster*. (We have now expanded the methods further as the reviewer suggested). With this approximation of nucleotide distribution we indeed see that the probabilities of the different asORFs are non-identical (Figures 1C and 1D). Overall, we agree that it is unlikely that in real genomes, asORF0 would be equally probable as asORF2 (we now explicitly state this in lines 166 – 169).

4. While an ORF's sequence and the biochemical properties of the encoded protein sequence, do depend on the codons other than start and stop, the presence and absence of an ORF (the questions pertaining to this study) only depend on these two kinds of codons. As we mentioned in the previous point, we do take into account the probability of the stop codons in the antisense frames as a function of the frequency of different codons in the sense ORF. Therefore, we believe this addresses the reviewer's criticism.

5. We respectfully disagree that our modeling approach (using different kinds of approximations) limits the usefulness of the study. All models are based on approximations (not only in biology), and should always be adapted to specific problems. We show that presence of an antisense overlap alone, given all other parameters be-

ing identical, can cause non-identical probabilities of asORFs and igORFs. This is an important result elucidating a fundamental property of the dsDNA based genome. Yes indeed, if we include further considerations, the results can change (as shown in Figures 1C and 1D) but the fundamental property still holds true.

C5) There is still not enough detail in this part of the methods to reproduce this analysis.

We have now expanded the section further. Here, we summarize the method briefly.

We count all the possible codons in a non-redundant list of ORF sequences. We divide the number of any one specific codon with the total number of codons found in these ORF sequences. This gives the codon frequency. Likewise for dicodon.

For intergenic trimers, we count all possible three letter words in the intergenic DNA. To this end use a sliding window approach (*e.g.* position 1 – 3, position 2 – 4, position 3 – 5). We divide the count of one specific trimer (such as ATG) with that of the total number of trimers, to obtain its frequency.

We have also included a script in our GitHub repo that does this calculation, and provide a brief description of the all the different scripts used in our analysis in a new supplementary section (9).

C6) I agree that using locus-specific parameters makes things more realistic. But that wasn't my point in this comment. Before comparing to the real data, the authors present the results of their model based on these two different types of parameters. My point was that both these non-locus-specific results cannot be true at the same time, and so a justification about why is expected. I fail to see this in the authors' response.

Perhaps we did not understand the reviewer's original comment correctly. We hope we have now understood what they are pointing at. We think that there is no one single truth that is highlighted by these results. For some cases the two approaches may yield similar results while in other cases they may yield very different results (as we can already see in case of the two organisms). In any case our models propose different null hypotheses (lines 419 – 427). It is difficult to state what the truth is without testing the hypotheses. When we compare the predictions with the data (Table 2), we find that the GC based model makes closer predictions to the observed value in some cases (asORF1 and igORF) whereas in the other cases, the oligomer based model makes closer predictions. Overall, if we take the sum of errors (normalized absolute difference) for all four ORFs into account, then the oligomer based model is 1.008 times better than the GC based model (that is, they are not very different).

However, in both models the observations were qualitatively similar to the predictions ($\text{asORF1} > \text{asORF2} \geq \text{asORF0}$). However, quantitatively the observed values are different from expected (asORF1 :more, asORF0 :less and igORF :less). Thus, a part of the observation can be explained by the null hypothesis. However, to state one truth, we could say that it would be unlikely that codons in a real ORF be based on uniform GC distribution. Hence, it is unlikely in real world for asORF0 and asORF2 to have identical probability (we now explicitly state this in lines 166 – 169). Apart from that we assume that the truth should come in various shades.

*C9) Figure 2 however is not huge, and the *D. melanogaster* results for B could easily be included to facilitate the comparison.*

We have expanded Figure 2 to include *D. melanogaster* results (panels D – F). Here we show unique ORFs representing the orthogroups. We have also updated the corresponding supplementary figure (Figure S2) where we show data from each specific *D. melanogaster* line, using the same kind of plots as in Figure 2.

C10) I'm glad the authors corrected the initial error. But in their reply the authors state that "predicted and observed values are now in agreement", contradicting their own text: "We found that the actual asORFs in the yeast genome were 1.6 – 24% fewer than expected". At the very least the comparison to the real data shows that the model is limited and this should be very clearly presented in the discussion. (it isn't currently)

Please also see our reply to **(C6)**. The model, that represents a null hypothesis, qualitatively agrees with the observation and hence partially explains it. It does not quantitatively agree in all the cases. We now highlight this point in lines 419 – 427. We now also state explicitly everywhere in the manuscript that the agreement between model and data is qualitative.

C11) I too believe that short ORFs are worth investigating; but that wasn't the comment. Given the model predictions, one would expect a sharper difference between the frames if the analyses were restricted only to longer ORFs. Why not test this directly when all the data are available to do so?

We have understood the reviewer's point now and have performed an analysis of longer ORFs from our sample. Generally, we detect a higher proportion of small asORFs in *D. melanogaster* (median ~ 22 codons) than in *S. cerevisiae* (median 25 codons). Because an ORF's probability reduces exponentially with its length, we used a cutoff of 90nt that is higher than 30nt but still allows a sizable number of

ORFs to be detected (Figure R1A). Both in *S. cerevisiae* and *D. melanogaster* we find that these long asORFs are more numerous in frame 1 than in the other two frames. While in *S. cerevisiae* these observations qualitatively agree with the predictions, *D. melanogaster* presents a more complex scenario. For small ORFs (< 25 codons), asORFs are predicted to occur the most likely in frame 2 and the least likely in frame 0 (Figure R1B). For longer ORFs, frame 0 has the highest likelihood of harboring an asORF (Figure 1D). However, we still find that the asORFs detected from our data are more frequently located in frame 1 irrespective of the minimum size cutoff. So why are asORFs more frequent in frame 1 than expected (this is the case in *S. cerevisiae* too, despite the qualitative agreement with the predictions)? So far the results suggest that the composition of known protein coding ORFs with overlapping antisense RNA, could be different from the averaged composition of all protein coding ORFs. To verify if this is the case, we measured the GC content of the *D. melanogaster* exons with an overlapping antisense RNA (wAS-exons) and compared them with that of all coding exons. We found that in each *D. melanogaster* line, wAS-exons had a significantly lower GC content (median ~0.41) relative to all exons (median ~0.45, one tailed Mann-Whitney U test, FDR adjusted $P < 10^{-16}$). We also found that the asORFs themselves were located in regions with an even lower GC content (median ~0.32, one tailed Mann-Whitney U test, FDR adjusted $P < 10^{-16}$). We found similar results with *S. cerevisiae* where the GC content of protein coding regions that overlap with an antisense RNA have a lower GC content (median 0.36) than all the protein coding regions in total (median 0.39, Mann-Whitney U test, $P < 10^{-16}$). This at least partially explains the high frequency of asORFs in frame 1 (Figure 1B) as their frequencies increase with decreasing GC content. We could not yet speculate the biological reason behind the low GC content of these regions. We now add this explanation in the manuscript (lines 248 – 261).

Figure R1: (A) Expected and observed number (vertical axis) of antisense ORFs in *S. cerevisiae*. (B) Probability of finding short ORFs in *D. melanogaster* based on nucleotide distribution estimated from oligomer frequencies.

C13) *The reply to C8 does not cover any deviation of the model from real data.*

This was the original **C13**:

I found the same issues in the fly comparison and, again, the results should have been presented in the main text. Here, an important deviation from the model is explained once more by “codon usage bias”. If the authors have in mind something more specific as to how codon usage bias accounts for longer ORF lengths in frame 0, then they should test it.

Our previous response to **C8** explains how codon usage differs between the two organisms that cause asORF-0 to be relatively more frequent (hence also longer) in *D. melanogaster* in comparison with *S. cerevisiae*. In short, two codons that code for leucine (TTA, CTA), one that codes for serine (TCA), encode a stop codon in the antisense frame 0. While the two organisms have similar frequencies of serine and leucine in their proteome, *D. melanogaster* uses TTA, CTA, and TCA more rarely than the other codons. This causes a rarity of stop codons in antisense frame 0, which not only increases the probability of finding these ORFs (original response in **C8**) but also increases the expected length of the ORFs (explanation for **C13**).

C14) *So from what I understand this is a totally different explanation from the initial one, but it is still a speculative one. My comment still holds: this is so central to the study’s message that it needs to be tested.*

First, we added a different explanation because it was more apt in our opinion. We now don’t think that pseudogenes and transposons may be the major factors responsible for the observed differences. However, they indeed do occur in intergenic regions and not in antisense overlapping regions.

Our explanation for more frequent igORF is that our analysis of asORF is very restrictive. We believe that most asORFs in *D. melanogaster* will have parts of their sequence that are overlapping and parts that are non-overlapping. This is because the sense codons contain introns. Introns can also change the frame of overlap such that different parts of an asORF can have different frames of overlap. Thus, we cannot categorize them into one of the three types. Hence we used asORFs that completely overlap with an exon.

While we think that ideally this should not cause a deviation from the deterministic prediction, it is possible that the assumptions for an ideal case are not fulfilled. First,

the much smaller sample size of asORFs relative to igORFs itself cause a large deviation from the ideal probability. For example, asORFs in *S. cerevisiae* are 24 times less numerous than igORFs. However, asORFs in *D. melanogaster* are not only ~ 2 times fewer than *S. cerevisiae* asORFs, but are also ~ 1990 times fewer than *D. melanogaster* igORFs. Second, the smaller search space for asORFs also forces the ORFs to be mostly small (median length < 24 codons, *i.e.* half the ORFs are smaller than this size; Figure S2). For small ORFs sizes (< 25 codons), igORFs can actually be more frequent than asORFs (Figure 1). Finally, there can be other idiosyncratic factors that we are not aware of (for example, the composition of the sense exons and the selective forces acting on them).

Overall, the deviation from the predictions could result due to a combination of these different factors. Therefore, we can at best be speculative. We have included these explanations in lines 241 – 247.

REVIEWERS' COMMENTS

Reviewer #2 (Remarks to the Author):

The authors have addressed some of my concerns and have maintained their arguments for others. I cannot say I fully understand the authors' response to some of my previous comments, like the first one, but that could also be my problem. I think we also have a difference of opinion on how results from a mathematical model should be perceived and compared to real data. But at this point I have no more comments and I feel it is time that this work reaches its audience.